# Periodic fasting and refeeding re-shapes lipid saturation, storage, and distribution in brown adipose tissue

Xing Zhang[1,2,3,4☯], Ting Jiang[5,6☯], Chunqing Wang[1☯], Valeria F. Montenegro Vazquez[2], Dandan Wu[7], Xin Yang[1], Que Le[1], Melody S. Sun[1], Xiaofei Wang[1], Xuexian O. Yang[2,7], Jing Pu[2,7], Matthew Campen[5,6], Changjian Feng[5,6*], Meilian Liu[1,2*]

1 Department of Biochemistry and Molecular Biology, University of New Mexico Health Sciences Center, Albuquerque, New Mexico, United States of America, 2 Autophagy Inflammation and Metabolism Center for Biomedical Research Excellence, University of New Mexico Health Sciences Center, Albuquerque, New Mexico, United States of America, 3 The National and Local Joint Engineering Laboratory of Animal Peptide Drug Development, College of Life Science, Hunan Normal University, Changsha, Hunan, China, 4 Hunan Provincial Key Laboratory of Animal Intestinal Function and Regulation, College of Life Science, Hunan Normal University, Changsha, Hunan, China, 5 Department of Pharmaceutical Science, College of Pharmacy, University of New Mexico Health Sciences Center, Albuquerque, New Mexico, United States of America, 6 Metals in Biology and Medicine Center for Biomedical Research Excellence, College of Pharmacy, University of New Mexico Health Sciences Center, Albuquerque, New Mexico, United States of America, 7 Department of Molecular Genetics and Microbiology, University of New Mexico Health Sciences Center, Albuquerque, New Mexico, United States of America

☯ These authors contributed equally to this work.
* cfeng@unm.edu (CF); meilianliu@salud.unm.edu (ML)

## Abstract

Brown adipose tissue (BAT) functions as a metabolic sink, efficiently processing fatty acids (FAs), glucose, and amino acids, playing a pivotal role in metabolic regulation and energy homeostasis. However, the metabolic adaptations enabling BAT to respond to fasting and refeeding cycles are not well understood. Using mass spectrometry techniques—Liquid Chromatography (LC), Capillary Electrophoresis (CE), and Spatially Resolved Imaging—we demonstrate that BAT exhibits a unique free fatty acid (FFA) and lipid-bound FA profile, with enrichment of very long-chain polyunsaturated fatty acids (VLC-PUFAs) and C13-C14 FAs compared to white adipose tissue (WAT) in male C57BL/6 mice. Alternate-day fasting (ADF) triggered a dynamic change of these FFAs in BAT, accompanied by selective alterations of upper glycolysis, glyceroneogenesis, and triglyceride synthesis, a shift less pronounced in WAT. Additionally, several BAT lipid species, including glycerolipids, glycerophospholipids, and sphingolipids, transitioned from highly unsaturated to more saturated lipids upon refeeding, alongside significant spatial and dynamic reprogramming. Mechanistically, periodic fasting and refeeding activated mTORC1, and genetic inactivation of mTORC1 in BAT diminished ADF-induced lipid saturation, storage, and redistribution in the C57BL/6 background. These findings reveal that while BAT generally prefers

**Data availability statement:** The raw metabolomics, lipidomics, and MALDI MSI data are available in MASSIVE datasets with identifier MSV000100122. All relevant data are within the paper and its Supporting information files.

**Funding:** This work is supported by National Institute of Health, including the National Institute of Diabetes and Digestive and Kidney Disease (R01 awards, DK110439 and R01DK132643, to M.L.), the National Institute of General Medical Science (S10OD032175 to C.F., P20 Award GM130422 to Matthew Campen, and P20 Award GM121176 to Vojo Deretic), the National Center for Advancing Translational Sciences (UM1TR005466 to Nancy Pandhi, Sally Radovick, and Matthew Campen and K12TR005467 to Judy Cannon and Jason Wertheim), the American Heart Association (Postdoc Fellowship Awards 20POST35120020 to X.Z, and the Hunan Normal University (Startup research grants 0531120 and 0531320 to X.Z). This project was also supported by the University of New Mexico School of Medicine (the Dedicated Health Research Funds to M.L).

**Competing interests:** The authors have declared that no competing interests exist.

**Abbreviations:** AAV, adeno-associated virus; AD, ad libitum; ADF, alternate-day fasting; BAT, Brown adipose tissue; Cers, ceramides; CE-TOFMS, Capillary Electrophoresis-Time-of-Flight Mass Spectrometry; DAGs, diacylglycerols; DHAP, dihydroxyacetone phosphate; DHB, 2,5-Dihydroxybenzoic acid; FAs, fatty acids; FFAs, free fatty acids; F1,6BP, fructose-1,6-bisphosphate; F6P, fructose-6-phosphate; GAP, glyceraldehyde-3-phosphate; GAPDH, glyceraldehyde 3-phosphate dehydrogenase; GPD1, glycerol-3-phosphate dehydrogenase; G3P, glycerol-3-phosphate; G6P, glucose-6-phosphate; HCA, hierarchical cluster analysis; HFD, high-fat diet; HK2, hexokinase; H&E, hematoxylin and eosin; IF, Intermittent fasting; LC, Liquid Chromatography; LCFAs, long-chain fatty acids; LC-TOFMS, Liquid Chromatography-Time-of-Flight Mass Spectrometry; LPC, lysophosphatidylcholine; MALDI-MSI, Matrix-Assisted Laser Desorption/Ionization mass spectrometry imaging; MG, monoacylglycerol; mTOR, mechanistic targets of rapamycin; PAS, Periodic Acid–Schiff; PCs, phosphatidylcholines; PCA, principal component analysis; PEs,

unsaturated fats, it undergoes substantial lipid saturation and spatially dynamic reprogramming in response to fasting and refeeding, offering new insights into BAT's adaptive role in metabolic homeostasis.

## Introduction

Brown adipose tissue (BAT), different from typical white adipose tissue (WAT), transfers chemical energy into heat and improves energy consumption, thereby combating obesity and metabolic disease. Upon activation, brown adipocytes within the BAT also function as an effective energy sink that disposes lipids, glucose, and amino acids, leading to a profound improvement in energy and glucose homeostasis [1–5]. While cold-induced lipolysis and fatty acid (FA) oxidation has been well established, the understanding of BAT metabolic adaption to fasting and refeeding remains largely unknown.

Intermittent fasting (IF), an eating pattern that cycles between periods of fasting and eating, has gained attention as an effective approach to improve obesity-induced metabolic dysfunction such as insulin resistance and type 2 diabetes [6–8]. IF has been shown to elicit its metabolic benefits by increasing WAT mitochondria protein content and browning of white fat, improving glucose tolerance and dyslipidemia in circulation [6–14]. Metabolic mediators such as gut fermentation products acetate and lactate as well as adipocyte-derived prostaglandins and cytokine such as VEGF mediate IF-elicited thermogenesis by targeting beige adipocytes and resident macrophages, respectively diabetes [6–8,14,15]. However, while BAT exhibits distinct phenotypes in response to IF diet regimen compared to WAT [13], how IF impacts BAT's metabolic sink property remains to be established.

The intracellular triglyceride (TG) via lipolysis is the primary energy source of BAT towards non-shivering thermogenesis upon sympathetic tone stimulation [16,17]. TG-rich lipoproteins, which deliver FAs as the major fuel, are also associated with cold-induced uncoupling process [18,19]. Under fasting conditions, lipoprotein-derived FAs become the primary fuel for BAT thermogenesis [20]. When lipolytic enzymes are deficient in brown adipocytes, BAT compensates by increasing the uptake and oxidation of circulating FAs, which substitute for intracellular TG stores to support cold-induced thermogenesis [21,22]. On the other hand, it has been increasingly recognized that glucose is used as a precursor for de novo lipogenesis, creating FAs for BAT oxidation [23]. The increase in glucose disposal also explains the property of BAT as a metabolic sink, contributing to systemic glucose homeostasis under such circumstances [4,24]. Within BAT, glycolysis has been linked to glycogenesis, pentose phosphate pathway (PPP), lactate synthesis, de novo lipogenesis, and glyceroneogenesis, all of which have certain connection with thermogenesis despite a controversy on whether glucose directly fuels thermogenesis [24–27]. In cold adaptation, BAT also dramatically increases nitrogen uptake by net consuming amino acids [23]. However, how BAT metabolically adapts to periodic fasting and refeeding remains largely unknown.

phosphatidylethanolamines; PEP, phosphoe-
nolpyruvate; PG, phosphatidylglycerol; PGK1,
phosphoglycerate kinase 1; PIs, phosphatidy-
linositols; PKM, pyruvate kinase; PPP, pentose
phosphate pathway; PS, phosphatidylserine;
PUFAs, polyunsaturated fatty acids; ROIs,
Regions of interest; RT, room temperature;
SM, sphingomyelins; SQDGs, sulfoquinovo-
syl diacylglycerols; TCA, tricarboxylic acid;
TEM, transmission electron microscopy; TG,
triglyceride; TPI, triosephosphate isomerase;
TPP, thiamine pyrophosphate; VLC-PUFAs, very
long-chain polyunsaturated fatty acids; WAT,
white adipose tissue; 2PG, 2-phosphoglycerate;
3PG, 3-phosphoglycerate.

The mechanistic targets of rapamycin (mTOR), an intracellular energy sensor that integrates distinct signals such as hormones, nutrients, and stress, is a vital regulator of multiple cellular processes such as protein translation, lipid metabolism, cell growth, and survival [28]. mTOR exists in two distinct complexes, mTORC1 and mTORC2, which differ in subunit compositions and biological function [29]. mTORC1 contains the rapamycin-associated TOR protein (Raptor), while mTORC2 contains the rapamycin-insensitive companion of mTOR (Rictor) [29]. Raptor is also a regulatory-associated protein of mTOR that binds to and mediates mTOR action [30]. Loss of raptor eliminates mTORC1 activity irrespective of feeding status [31]. mTORC1 is understood to regulate multiple metabolic processes including protein synthesis, lipogenesis, energy expenditure, and autophagy [28,32] and is highly active in the adipose tissue of obese and high-fat diet (HFD)-fed rodents [33,34]. Furthermore, excessive mTORC1 signaling results in the development of adipos-ity and obesity [32,35–38]. However, accumulated evidence has raised questions regarding its role in AT thermogenesis [35,36,39–45]. Several studies agree that adipose inhibition of mTORC1 leads to browning of WAT and that aberrant mTORC1 signaling in AT suppresses thermogenesis and exacerbates diet-induced obesity and insulin resistance in mice [35,36,39,40]. However, other studies report that suppres-sion of mTORC1 signaling in AT results in impaired thermogenesis [41,42]. While the work from Lee, Shan, and colleagues suggests an inhibitory effect of mTOR signaling on WAT browning, both studies argue that adipose-specific inhibition of mTOR or mTORC1 results in lipodystrophy and systemic insulin resistance during postnatal growth [43–45]. However, the development-independent effects of mTORC1 on BAT metabolic adaptation remains incompletely understood.

Using Liquid Chromatography (LC), Capillary Electrophoresis (CE), and Matrix-Assisted Laser Desorption/Ionization mass spectrometry imaging (MALDI-MSI), we demonstrated that BAT is characterized by a higher degree of unsaturated FAs and glycolipids compared to white fat tissues. Periodic fasting and refeeding rapidly trigger a shift in lipid composition from high to low degrees of unsaturation, reshap-ing BAT's lipid saturation, diversity, and spatial distribution. Additionally, elevated mTORC1 signaling partially mediates the refeeding-induced shift from high to low unsaturation in lipids, promoting TG biosynthesis and BAT whitening. Our study provides crucial insights into BAT lipid heterogeneity and diversity, emphasizing its significant role in IF-related metabolic adaptations and highlighting its potential thera-peutic applications.

## Materials and methods

### Materials

Antibodies against HSL (4107S), phospho-HSL at Ser563 (4139S), phospho-S6K at Thr389 (9205L), phospho-AKT at Thr308 (9257S), phospho-AKT at S473 (4051S), AKT(9272S), and RAPTOR (2280S) were from Cell Signaling Technol-ogy. SCD1(A16429), SLC27A1(FATP-1, A12847), MLXIPL(ChREBP, A7630), ELOVL6(A21094), SCD5(A13127), and CD36(A5792) were from ABclonal Science

Anti-CD31 was from R&D systems (AF3628). Anti-S6K (CG1396-100) was from Cell Applications. Anti-UCP1 (ab23841) and PRDM16 (ab106410) were from Abcam. Anti-PGC1α (ST1204-100UL) was purchased from MilliporeSigma. Anti-C/EBPβ (sc-150) was from Santa Cruz Biotechnology. Prestained Protein Ladder (26616), Anti-TUBB (tubulin beta class 1) was from Sigma-Aldrich (T8328).

## Animals

*Raptor* floxed (Stock number: 13188) and C57BL/6 mice (Stock number: 000664) were purchased from Jackson Laboratory as we described previously [15]. Ten-week-old *Raptor* floxed mice underwent surgery for intra-BAT administration. A longitudinal incision at the interscapular region was performed, followed with six injections with 10 µl of $10^{13}$ vg/ml adeno-associated virus (AAV) that encoded mouse Cre (VectorBuilder Inc, Chicago, IL) into brown fat depot. Mice were followed with alternate-day fasting (ADF) for 3 cycles 2 weeks postvirus injection. All animals were housed in a pathogen-free barrier facility with a 12 h light/12 h dark cycle with food and water *ad libitum* (AD). All animal experimental protocols assigned under protocol number 25-201670-HSC were reviewed and approved by the Animal Care Committee of the University of New Mexico Health Sciences Center. All animal procedures complied with the U.S. Animal Welfare Assurance # D16-00228 (A3350-01), the Public Health Service Policy on Humane Care and Use of Laboratory Animals, and adhered to the Guide for the Care and Use of Laboratory Animals (National Research Council, 8th edition).

## Intermittent fasting

Ten-week-old male C57/BL6 mice randomly underwent AD and ADF ended with fasting (Fas) or refeeding (Fed) under room temperature (RT; 22 ℃) or Thermoneutrality condition (30 ℃). The AD group of mice were given unrestricted access to the diet, while ADF group of mice were fasted for 24 hours followed with refeeding for alternating 24-hour periods of free access to food. Mice were euthanized 24 hours postfeeding (AD), refeeding (Fed), or fasting (Fas).

## Metabolomics and lipidomic analyses

Capillary Electrophoresis-Time-of-Flight Mass Spectrometry (CE-TOFMS) analysis was used to profile hydrophilic and charged metabolites. In brief, fat tissue samples were mixed with 50% acetonitrile in water (v/v) containing internal standards (10 µM) and homogenized with a homogenizer (3,500 rpm, 60 s × 9 times). The supernatant (400 µL) was then filtrated through a 5-kDa cut-off filter (ULTRAFREE-MC-PLHCC, Human Metabolome Technologies, Yamagata, Japan) to remove macromolecules. The filtrate was centrifugally concentrated and resuspended in 25 µL of ultrapure water immediately before the measurement. The compounds were measured in the Cation and Anion modes of CE-TOFMS-based metabolome analysis. The samples were diluted for the measurement to improve analysis qualities of the CE-MS analysis.

Liquid Chromatography-Time-of-Flight Mass Spectrometry (LC-TOFMS) was used for metabolic profiling of more hydrophobic and lipid metabolites. The fat tissue samples were mixed with 500 µL of 1% formic acid in acetonitrile (v/v) containing internal standards (10 µM), homogenized with a homogenizer (1,500 rpm, 120 s × 2 times). The mixture was homogenized again after adding 167 µL of Milli-Q water and then centrifuged (2,300 *g*, 4 ℃, 5 min). After the supernatant was collected, 500 µL of 1% formic acid in acetonitrile (v/v) and 167 µL of MilliQ-water were added to the precipitation. The homogenization and centrifugation were performed as described previously, and the supernatant was mixed with the previously collected one. The mixed supernatant was filtrated through a 3-kDa cut-off filter (NANOCEP 3K OMEGA, PALL Corporation, Michigan, USA) to remove proteins and further filtrated through a column (Hybrid SPE phospholipid 55261-U, Supelco, Bellefonte, PA, USA) to remove phospholipids. The filtrate was desiccated and re-suspended in 200 µL of 50% isopropanol in Milli-Q water (v/v) immediately before the measurement. The compounds were measured in the Positive and Negative modes of LC-TOFMS-based metabolome analysis. The samples were diluted for the measurement to improve analysis qualities of the CE-MS analysis.

For both CE-TOFMS and LC-TOFMS measurements, the detected peaks were annotated according to HMT's standard library and Known-Unknown peak library. The results of principal component analysis (PCA) used the detected peaks, while the hierarchical cluster analysis (HCA) results are shown as a HeatMap.

## MALDI-MSI analysis

Interscapular BAT depot was carefully dissected with particular attention to removing surrounding WAT, and half of the total BAT tissue (one side of butterfly) were used for MALDI. Additionally, the center region of BAT tissue was screened using MALDI. For MSI sample preparation, freshly harvested mouse brown fat tissues were embedded in M-1 embedding matrix, frozen in liquid nitrogen, and sectioned at 10 µm thickness. A total of nine sections (representing 3 biological replicates from 3 experimental groups) were mounted on a single Bruker Intellislide. The slide was dried in a desiccator for 0.5 hour prior to matrix application. A matrix solution of 2,5-Dihydroxybenzoic acid (DHB, Sigma-Aldrich) was prepared in 900:100:1 acetonitrile:water:TFA mixture at a concentration of 15 mg/ml. The DHB solution was sprayed using an HTX M3+matrix sprayer (HTX Technologies, Chapel Hill, NC) with the following settings: temperature = 60 °C, pressure = 10 psi, flow rate = 125 µL/min, velocity = 1,200 mm/min, track spacing = 3 mm, number of passes = 14, pattern = CC, and drying time = 10 s.

For MSI data acquisition, data were acquired on a timsTOF Flex Maldi 2 instrument (Bruker Daltonics, Bremen, Germany) in tims on positive ion mode. Prior to data acquisition, both the $m/z$ and mobility were calibrated using the ESI L Low Concentration Tuning Mix (Agilent Technologies, Santa Clara, CA). Acquisition settings included an ion transfer time of 85 µs, a prepulse storage time of 10 µs, collision RF of 1800 Vpp, TIMS funnel 1 RF at 500 Vpp, TIMS funnel 2 RF at 350 Vpp, multipole RF at 300 Vpp, and a mass range of 300–1,350 $m/z$. The mobility range was set with a 1/k0 start of 0.7 V·s/cm², and 1/k0 end of 1.8 V·s/cm², with a ramp time of 200 ms. Imaging was performed at a spatial resolution of 20 µm with beam scan on (16 µm), using 1 burst of 200 shots per pixel, at 70% laser power, and a 10 kHz laser repetition rate. In Fig 5, the pixel numbers are 27845, 13661, and 20297 for AD, 21155, 24338, and 32046 for fasting, and 13283, 8145, 10476 for Refed group. In Fig 7, the pixel numbers are 27757, 22415, 11788, and 11130 for control AD, 20954, 15890, 13095, and 11342 for control Refed, 16481, 16510, 9544, and 17956 for KO AD, and 10278, 5776, 8835, 10745, and 10712 for KO Refed. For MSI data analysis, imaging data processing and analysis were conducted using SCiLS Lab (v.2024b core, Bruker Daltonics, Bremen, Germany). Feature findings were performed with a 15 ppm interval width and total ion count was used for normalization. Regions of interest (ROIs) were defined in SCiLS Lab by segmentation using the bisecting k-Means algorithm without denoising and correlation distance was selected as metric. The ROIs were validated by comparison to hematoxylin and eosin (H&E)-stained images of the same tissue sections, which were washed and stained after the imaging experiment. The ROI with the untargeted feature list from feature finding was then imported into MetaboScape (v.2024, Bruker Daltonics, Bremen, Germany) for lipid annotation. Lipid searches were performed against the default lipid database using a mass tolerance of ±0.01 $m/z$ for positive polarity ([M+H]⁺, [M+Na]⁺, [M+K]⁺). Lipid identification was based on mass accuracy, isotopic pattern, and mobility, with annotation tolerances set to: 2 ppm (narrow) and 5 ppm (wide) for $m/z$, 50 (narrow) and 250 (wide) for mSigma, and 1% (narrow) and 3% (wide) for CCS. The tentatively identified feature list was exported from MetaboScape and re-imported to SCiLS Lab to generate ion images of features of interest. The average abundances of features of interests in the ROI were then compared across the three experimental groups.

## Glycogen staining

For glycogen staining, the Periodic Acid–Schiff (PAS) kit (Sigma #395B) was used, and sections were treated following the manufacturer's instructions. In brief, adipose tissue samples were fixed in 4% paraformaldehyde at RT overnight. Paraffin-embedded tissues were de-paraffinized for 3 times in xylene and subsequently rehydrated from serial ethanol (100%

ethanol, 95% ethanol, 85% ethanol, or 70% ethanol) to ddH$_2$O, and then oxidized in 0.5% periodic acid solution for 5 min followed with Schiff reagent for 15 min. After dehydration, the staining samples with coverslip were used for the image.

### Transmission electron microscopy (TEM)

For in vivo study, brown fat samples were collected from 10-week-old male C57/BL6 mice. Tissue samples were fixed with 0.1 M sodium cacodylate buffer containing 3% formaldehyde and 2% glutaraldehyde for 1 h. Tissue samples were then washed with cacodylate buffer followed by secondary fixation in 1% osmium-1.5% potassium ferricyanide at RT for 1 h. Sample was then infiltrated with Epon-Araldite resin, embedded, and heat-cured. The coverslips were removed from the cured resin blocks, and the pieces from the resin blocks were re-mounted for en face sectioning. Thin sections (~80 nm) were mounted on Cu grids, poststained with uranyl acetate and Reynold's lead citrate, and examined in a Hitachi HT7700 TEM at 80 kV. Images were captured with an AMT XR-81 CCD camera, and a total of 25 cells/slide underwent imaging. 3 images/cell were taken. For the quantification, the total number of mitochondria in the defined areas (272,475 µm$^2$)/cell were counted and analyzed using the average number in each area. The number of mitochondria was expressed as the number of mitochondria and lysosomes per nucleus as described previously [46].

### Energy expenditure

Ten-week-old male C57/BL6 mice were individually housed in eight separate Promethion Metabolic Phenotyping Systems (Sable Systems International) coupled with a temperature-controlled chamber. And the mice randomly underwent AD and ADF ended with fasting (Fas) or refeeding (Refed). Oxygen consumption (VO$_2$), carbon dioxide release (VCO$_2$), food intake, water intake, and the activity of each animal were monitored at room temperature (22 °C) for 3 cycles of ADF. The data were analyzed using CalR (version 1.3; https://calrapp.org/) [47,48].

### Hematoxylin and eosin staining and immunostaining

For H&E staining, adipose tissues from 10-week-old mice were fixed with 4% formalin for 48 h and embedded in paraffin. Tissue sections (5-µm thick) were stained with H&E according to standard protocols then analyzed using the NIH ImageJ software. For the immunofluorescence staining, tissue slides were stained with primary antibodies against CD31 (1:500 dilution), UCP1 (1:500 dilution) and followed with second antibody, and then mounted. The images were captured using the EVOS FL Cell Imaging System.

### RNA extraction and real-time PCR

Total RNA was extracted from tissue samples using the PureLink RNA Mini Kit (12183025 Thermo Fisher). The purity and concentration of total RNA were determined by a Nano Drop spectrophotometer (Thermo Fisher). 1 µg of total RNA was reverse-transcribed using High-Capacity cDNA Reverse Transcription kit (4368813 Thermo Fisher). Real-time PCR amplification was detected using SYBR Green PCR master mixture (330502 Qiagen) on a Roche 480 Real-time PCR system. Primer sequences are listed in S1 Table. Each pair of primers amplified products spanning several exons, distinguishing spliced mRNA from genomic DNA contamination. The reaction yielded about 200 bp of PCR products for the specified genes and internal control genes. The relative expression levels of target genes were normalized to 36b4.

### Statistics

HCA and PCA were performed using statistical analysis software (developed at HMT , Tsuruoka, Japan) in the studies of metabolomics and lipidomics. Statistical analysis of the rest of data was performed using a two-tailed Student $t$ test between two groups or one-way ANOVA among three diverse groups. All the results were presented as the mean ± S.E.M., and $P$-value of <0.05 was statistically significant.

## Results

### Periodic fasting and refeeding differentially reprograms BAT and WAT metabolism

IF has been gaining increased attention as an effective approach for weight loss and improvement of insulin sensitivity [6–11,14]. However, how various fat depots adapt to acute IF remains incompletely understood. To this end, 3-month-old male C57BL/6 mice underwent 3 cycles of periodic fasting and refeeding (S1A Fig). There was no significant difference in the body mass between AD and refeeding (Refed) groups, while food intake was significantly higher in ADF at refeeding stage compared to AD group (S1B and S1C Fig). Consistent with this, gWAT, iWAT, and liver became shrunken during fasting but restored by refeeding (Figs 1A, 1B, and S1D and S1 Table). However, BAT displayed a notable whitening upon refeeding accompanied with an increase in BAT mass and large lipid droplets despite mild changes during fasting compared to AD group (Figs 1A, 1B, and S1D), suggesting the distinct metabolic reprogramming of various fat depots during ADF. To further explore this, we performed lipidomic and metabolomic analyses of BAT and gWAT. The metabolites in the mouse BAT (B-AD, B-Fas, and B-Refed) and gWAT (g-AD, g-Fas, and g-Refed) from a total of six groups (3 samples/group, a total 18 tissue samples), were analyzed using CE-TOFMS and LC-TOFMS in two modes for cationic and anionic metabolites (HMT Metabolomics Technology, Boston, MA) (Fig 1C). CE-TOFMS was used to analyze 359 metabolites (212 metabolites in cation mode and 147 metabolites in anion mode) as shown by HMT's standard library, and LC-TOFMS was used to analyze 165 metabolites (93 metabolites in positive mode and 72 metabolites in negative mode). Based on the metabolite profiling, BAT and gWAT exhibited distinct metabolite profiles, and the AD, Fas, and Fed groups were well separated except for one sample of g-Fas group (Fig 1D and S2 Table).

In general, there were several obvious differences in metabolic pathways between BAT and gWAT (Fig 1E and S2 Table): (1) Compared to gWAT, BAT exhibits a higher abundance of most captured metabolites across various metabolic pathways; (2) There is a distinct FA profile between the two fat depots (S2 Table); (3) Upon refeeding, BAT underwent reprograming in multiple pathways, with notable elevation of those metabolites that belong to glycolysis, glyconeogenesis, glycogenesis, TG biosynthesis, nucleotide metabolism, glutamate metabolism, urea cycle, choline metabolism, FA oxidation, tricarboxylic acid (TCA) cycle, and mitochondria respiration, albeit to a much lesser extent in gWAT, and (4) gWAT exhibited a marked increase in a variety of microbial metabolites and dipeptides, which was not the case for BAT (S2 Fig).

### Very long-chain polyunsaturated fatty acids (VLC-PUFAs) and 13/14-carbon FAs are featured in BAT and reshaped by periodic fasting and refeeding

The lipidomic assays revealed that BAT and gWAT have their favorable free fatty acids (FFAs) at basal levels (Fig 1E and S2 Table). Among 60 targeted FFAs varying in carbon length and degrees of saturation, 12 of them were enriched in BAT but not in gWAT, including seven VLC-PUFAs (≥20). They were FA (24:5), FA (24:4), cis-4,7,10,13,16, 19-docosahexaenoic acid-2 (22:6, DHA), FA (22:5)-1, FA (22:4)-1, FA (22:3)-1, and arachidonic acid (20:4) (Fig 2A–2F). These VLC-PUFAs were elevated during fasting in both BAT and gWAT but markedly decreased by refeeding, dropping to levels even lower than those seen under AD conditions (Fig 2A–2F). In addition, BAT also favors those long-chain fatty acids (LCFAs) containing 13 and 14 carbons. Tridecanoic acid (13:0), myristic acid (14:0), myristoleic acid (14:1), and FA (14:1)-1, with higher degrees of saturation, were also higher in BAT than in gWAT under AD conditions and were further enhanced by fasting, while refeeding reversed these effects (Fig 2A and 2G–2J). BAT also expressed higher levels of cholesterol, 7-dehydrocholesterol-1, and 7-dehydrocholesterol-3, except for 22-hydroxycholesterol (S3A–S3D Fig). Additionally, bioactive lipids such as anandamide, oleoyl ethanolamide, palmitolethanolamide, and stearoyl ethanolamide were highly expressed in BAT but not in gWAT under both AD and fasting conditions (S3E–S3H Fig). These results further indicate a distinct profile of FFAs within BAT compared to WAT.

In contrast, some FFAs such as C15-19 LCFAs exhibited a lower abundance in BAT compared to gWAT (Fig 3). These LCFAs included FA (15:0), palmitic acid (16:0), FA (16:2)-1, FA (17:1), FA (17:2), oleic acid (18:1), linoleic acid (18:2),

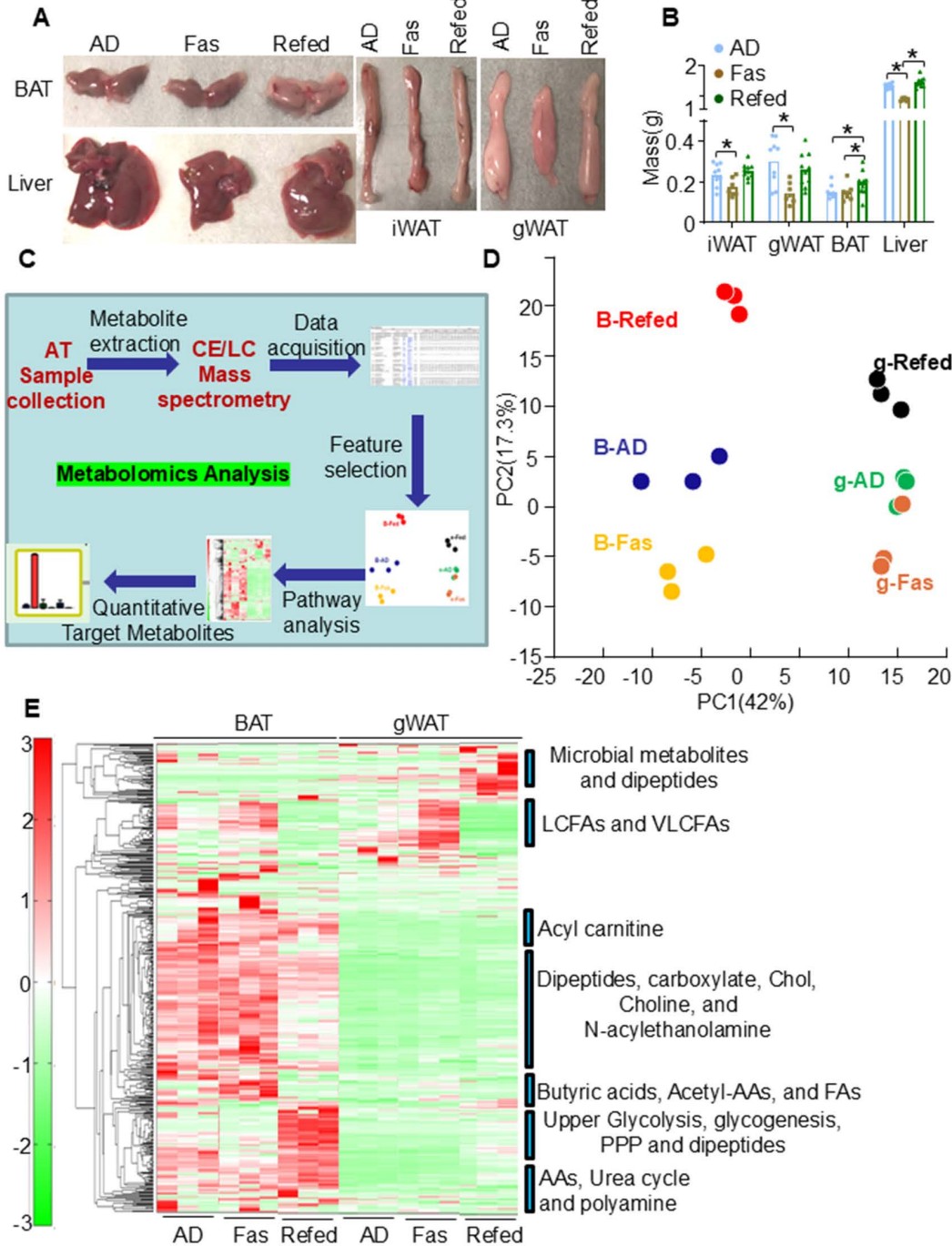

**Fig 1. ADF differentially reprograms metabolism in BAT and gWAT. A.** Representative images of gWAT, iWAT, BAT, and liver in three groups ad libitum (AD), fasting of ADF (Fas), and refeeding of ADF (Refed) ($n=8$/group). **B**. The mass of various fat tissues and liver after periodic fasting and refeeding ($n=8$/group). **C.** Workflow of CE-TOFMS and LC-TOFMS Analysis for adipose tissue samples. **D.** Principal component analysis (PCA) of metabolomics and lipidomics results using the detected peaks, PC1, the first principal component; PC2, the second principal component. The number in parentheses is the contribution rate, and the plot labels are sample names. **E.** Heatmap of adipose tissue metabolites with hierarchical cluster analysis (HCA) in BAT and gWAT from AD, Fas, and Refed groups. The raw data for 1B is provided in S1 Table, and the raw data for metabolomics and lipidomics analyses are presented in S2 Table. Data in Fig 1B is presented as mean ±SEM. *T* Test was used to analyze the data. *$P<0.05$.

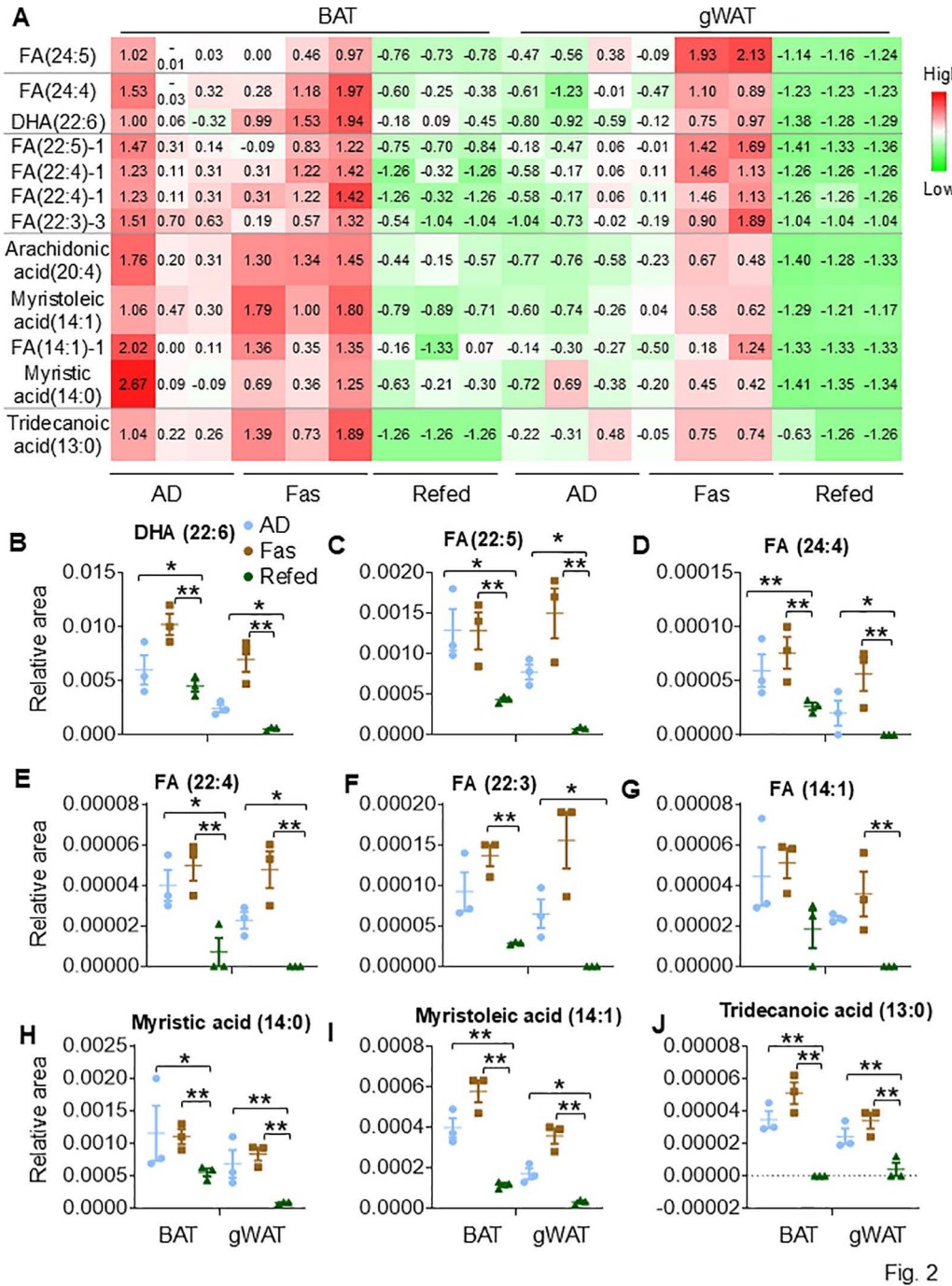

**Fig 2. VLC-PUFAs and C13-14 FFAs are highly abundant in BAT, which are decreased by refeeding. A.** Specific FFAs, such as VLC-PUFAs and C13/14 FAs, are enriched in BAT compared with gWAT during ADF. The data in 2A were quantified for individual FFAs, including *cis*-4,7,10,13,16, 19-docosahexaenoic acid-2 (22:6, DHA) **(B)**, FA (22:5) **(C)**, FA (24:4) **(D)**, FA (22:4) **(E)**, FA (22:3) **(F)**, FA (14:1) **(G)**, Myristic acid (14:0) **(H)**, Myristoleic acid (14:1) **(I)**, Tridecanoic acid (13:0) **(J)**. The raw data for metabolomics and lipidomics analyses are presented in S2 Table. Fig 2B–2J were analyzed by *T* Test. The data are presented as mean±SEM. *$P < 0.05$, **$P < 0.01$.

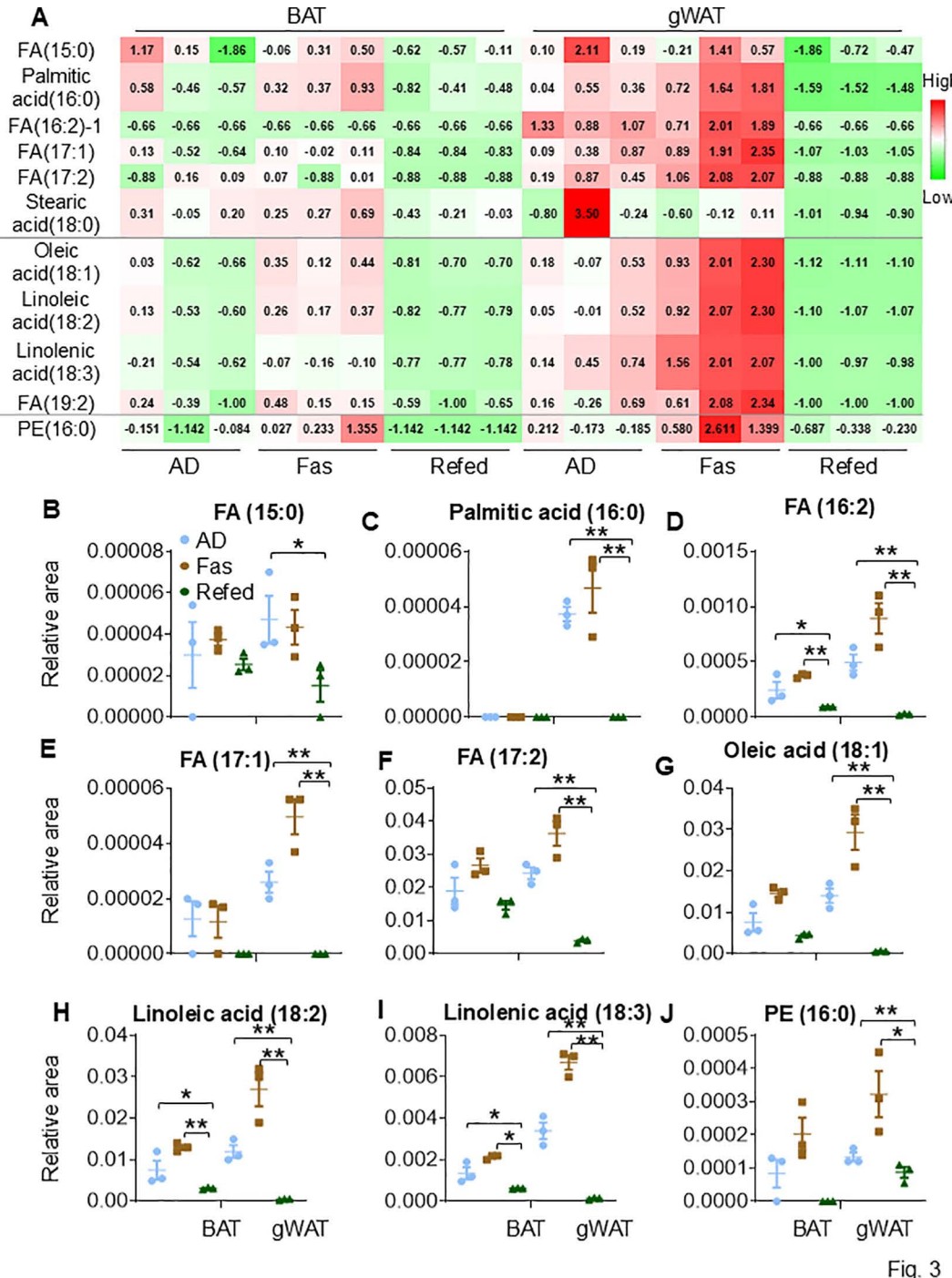

**Fig 3. Long-chain fatty acids (LCFAs) exhibit a lower degree of unsaturation as well as lower abundance in BAT compared to gWAT. A.** The table summarized representative 12 LCFAs that exhibit a lower degree of unsaturation in BAT and gWAT during ADF. The data in 2A were quantified for individual FFAs, including FA (15:0) **(B)**, Palmitic acid (16:0) **(C)**; FA (16:2) **(D)**, FA (17:1) **(E)**;, FA(17:2) **(F)**, Oleic acid (18:1) **(G)**, Linoleic acid (18:2) **(H)**, Linolenic acid (18:3) **(I)**, 1,2-Dipalmitoyl-glycero-3-phosphoethanolamine-1 (PE 16:0) **(J)**. The raw data for metabolomics and lipidomics analyses (3A-3J) are presented in S2 Table. Fig 3B–3J were analyzed by $T$ Test. The data are presented as mean $\pm$ SEM. *$P < 0.05$, **$P < 0.01$.

linolenic acid (18:3), and FA (19:2) (Fig 3A–3J). Notably, no significant difference was observed in the levels of stearic acid (18:0) between gWAT and BAT (Fig 3A–3I). These LCFAs were more prevalent in gWAT and increased by fasting, which was completely reversed by refeeding (Fig 3A–3I), with a similar but less pronounced trend seen in BAT (Fig 3A–3I). Consistently, 1,2-dipalmitoyl-glycero-3-phosphoethanolamine-1 (PE 16:0) was found at lower levels in BAT compared to gWAT under both AD and fasting conditions (Fig 3J). These findings highlight the distinct FFA diversity and unsaturation degree between the two fat depots and the IF-elicited changes in FFA saturation and dynamics. Whereas other targeted lipid categories, such as sphingolipids, prostaglandins, anandamides, and other phospholipids, were not captured in this analysis.

## Periodic fasting and refeeding elicits a decrease in highly unsaturated lipids, while increasing more saturated lipids

To further investigate various BAT lipid species, we performed MALDI-MSI, a powerful technique increasingly used to analyze complex lipid mixtures [49]. BAT lipids exhibited significant diversity, varying in headgroups, the polar chain-glycerol linkages, and the fatty acyl residues, including distinctions in double bond number and positions (S3 Table and S1 Raw Images). MALDI analysis revealed a broad range of lipid species, including ceramides (Cers), sphingomyelins (SM), diacylglycerols (DAGs), lysophosphatidylcholine (LPC), monoacylglycerol (MG), phosphatidylcholine (PC), phosphatidylethanolamine (PE), phosphatidylglycerol (PG), phosphatidylinositols (PIs), phosphatidylserine (PS), and TG, as well as modified lipids such as ceramide (HexCer), alkyl-PCs (PC (O)), phosphatidylinositol-ceramide (PI-Cer), and sulfoquinovosyl diacylglycerols (SQDGs) (S3 Table and S1 Raw Images). Consistent with BAT FFA profile, basal levels of polyunsaturated fatty acids (PUFAs)-enriched lipids such as LPC (20:4) and PC (38:5) (Fig 4A–4G), as well as HexCer (38:4:O2 and 40:4:O2), PCs (36:4, 38:4, 38:5, and 40:7), PI-Cers (34:4:O3, 36:6:O3 and 36:8:O3), PSs (36:2, 38:2, 38:8, and 40:5), and TG (54:5 and 56:4)-were relatively high in BAT (S3 Table and S1 Raw Images). These highly unsaturated lipids showed robust redistribution during fasting followed by a notable decrease in Refed BAT (Fig 4A–4D and S3 Table and S1 Raw Images). Conversely, refeeding markedly increased levels of lipids with a higher degree of saturation such as LPC (16:0) and PC (30:1), which were initially lower in BAT (Fig 4A and 4E–4G). Similar changes occurred on other lipids, including LPCs (16:1, and 18:0), PC (32:0, 34:3, 36:1, and 38:1), PSs (PS 30:1, 30:2, and 32:1) and TG (48:1, 50:1, and 50:2) (Fig 4A, 4E–4G and S3 Table and S1 Raw Images). This shift reflects a diet-induced transition of FA saturation levels in BAT. Additionally, regardless of fatty acyl composition, HexCers (28:6:O2 and 32:6:O3), MGs (18:3, 20:1, 20;4, and 22:4), and PEs (34:0, 34:2, 36:1, 38:1, 38:3, and 38:6) were significantly elevated in Refed BAT, with exception of PE-38:2, which showed an opposite trend (S3 Table and S1 Raw Images). On the other hand, Cers, DAGs, alkyl-PCs (PC (O)), PIs, PGs, SM, and SQDGs in BAT remained largely unaffected despite high PG-34:0 and SM-44:3:O2 level during fasting (S3 Table and S1 Raw Images). These findings highlight how periodic fasting and refeeding influence lipid saturation, diversity, and dynamics in BAT.

## BAT lipid distribution exhibits spatial heterogeneity and dynamics during periodic fasting and refeeding

Beyond the dynamics and saturation, BAT lipid distribution was found to be spatially heterogenous (Fig 4B, 4C, 4E, and 4F). At baseline, TG and MG levels were lower than those of glycerophospholipids and sphingolipids (S3 Table and S1 Raw Images). Refed BAT showed increased MGs and highly saturated TGs with notable spatial variability, showing elevated levels in some specific areas (Fig 5A), which support its whitening (Fig 1A). Refeeding stimulated the synthesis of glycerophospholipids and sphingolipids, in particular HexCers and more saturated LPCs, PCs, and PSs, with certain areas exhibiting dramatically higher levels than others, forming "sparkle" regions (Fig 5B–5D). As the segmentation showed, this localized stimulation (red) was associated with angiogenesis (blue) (Fig 5C) and vasculature zones (CD31 and H&E staining) (S3I, S3J, and S4A–S4C Fig), but not UCP1-high adipocytes (S4D Fig), suggesting the possible esterification occurred on newly available FAs entered adipocytes during refeeding. In contrast, highly unsaturated PCs, PSs, and PI-Cer showed higher basal levels and were more evenly distributed at baseline (Fig 5E). These phospholipids

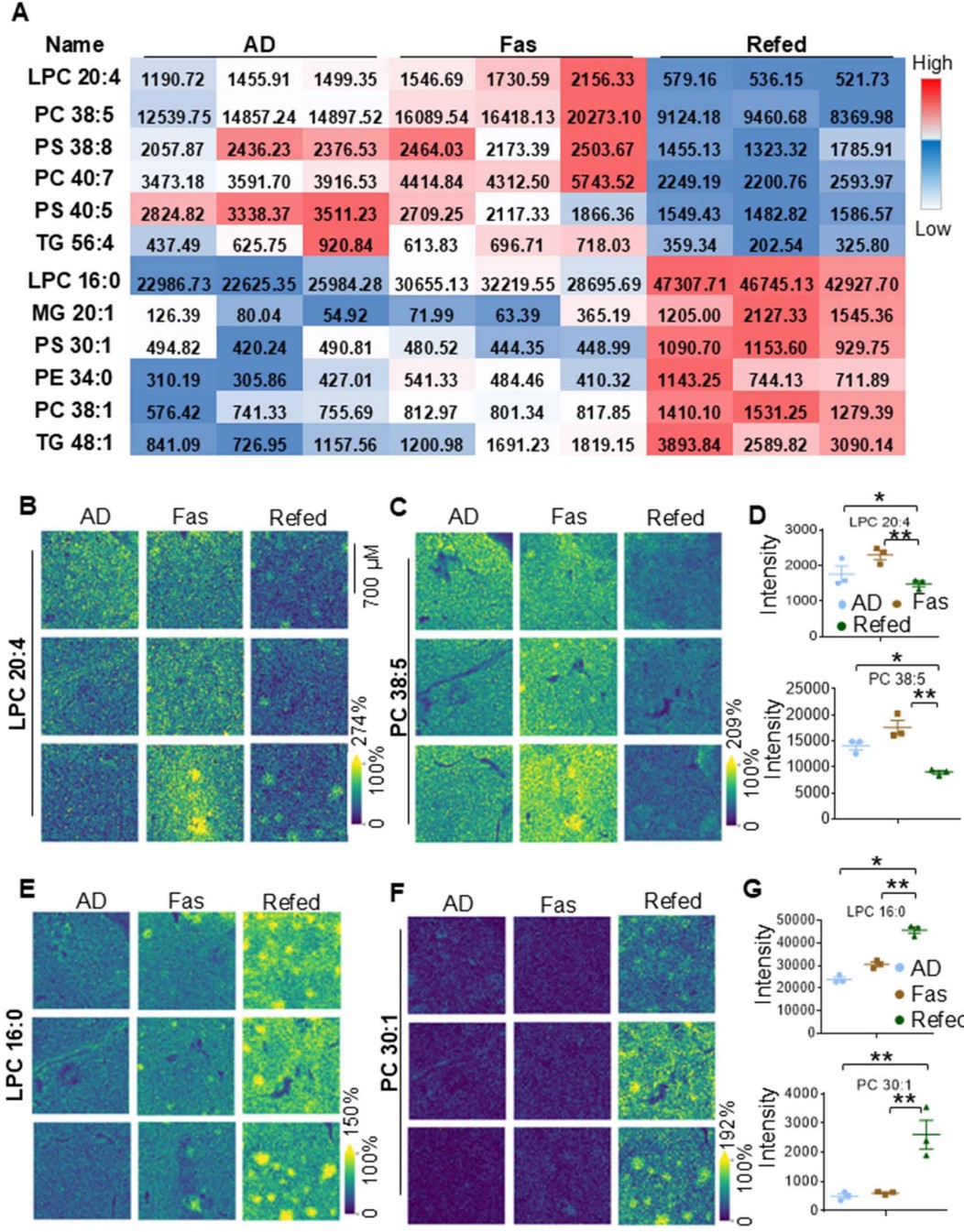

**Fig 4. Periodic fasting and refeeding elicits a shift of BAT lipids from high to low unsaturation. A.** The table summarizes representative glycero-lipids and glycerophospholipids that contain various degrees of unsaturation during ADF. The MSI showed a decrease for LPC 20:4 **(B)** and PC 38:5 **(C)** in BAT upon refeeding. **D.** The quantification of LPC 20:4 and PC 38:5 from Fig 4B and 4C. The MSI showed an increase for LPC 16:0 **(E)** and PC 30:1 **(F)** in BAT upon refeeding. **G.** The quantification of LPC 20:4 and PC 38:5 from Fig 4E and 4F. The raw data for MALDI analyses (4A–4G) are presented in S3 Table and S1 Raw Images. Data in Fig 4D and 4G are presented as the mean ± SEM. *$P < 0.05$; **$P < 0.01$.

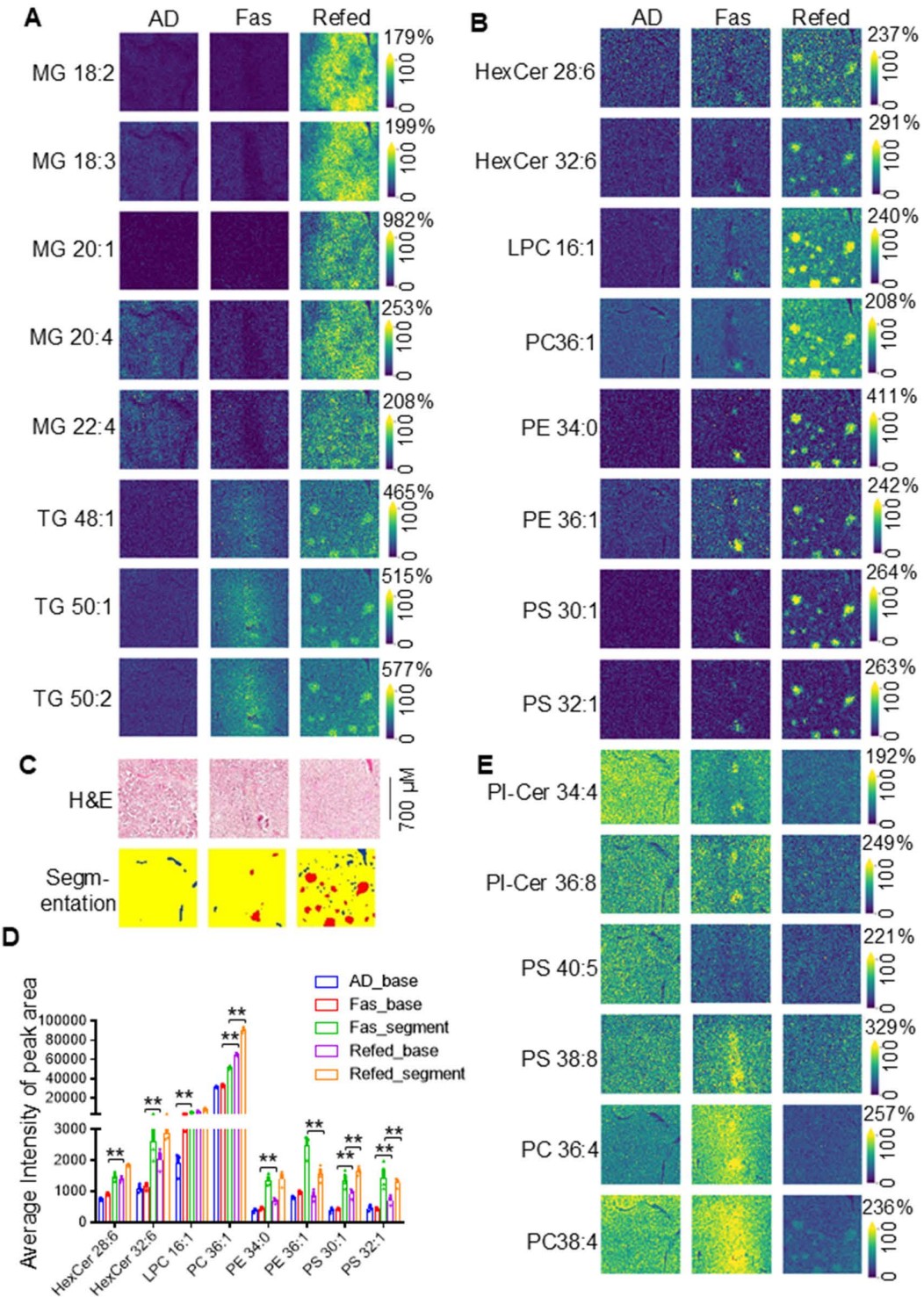

**Fig 5. BAT lipid distribution shows spatial heterogeneity and dynamics during periodic fasting and refeeding. A.** Representative images of MGs and highly saturated TGs in BAT during ADF. **B.** The Representative images of hexosylceremide and glycerophospholipids in BAT during ADF. **C.** H&E imaging and segmentation of MSI of BAT. **D.** Quantification of ion content peak area of segmented droplets and the rest areas. **E.** High-degree unsaturated PCs, PSs, and PI-Cer showed higher basal levels in AD, which was redistributed by fasting followed with a notable reduction upon refeeding. The raw data for 5D is provided in S1 Table, and the raw data for MALDI analyses (5D and 5E) are presented in S3 Table and S1Raw Image was analyzed by T test and the data are as the mean±SEM. *P<0.05.

exhibited a distribution variability under fasting followed with a significant reduction during refeeding (Fig 5E), suggesting the shift of lipids from high to low unsaturation. Their spatial heterogeneity mirrored that of MG and saturated TGs during fasting, followed by an overall decrease upon refeeding (Fig 5E and S3 Table). These findings reveal complex spatial heterogeneity and dynamic patterns across BAT lipid species, including sphingolipids, glycolipids, and phospholipids.

## Refed BAT is highly active in upper glycolysis, glyconeogenesis, and TG synthesis

As expected, both BAT and gWAT displayed upregulation of metabolites in glycolysis upon refeeding (Fig 1E and S2 Table). However, Refed BAT demonstrated a marked increase in the upper glycolysis, but not the lower glycolysis, as evidenced by elevated levels of intermediate metabolites, including glucose-6-phosphate (G6P), fructose-6-phosphate (F6P), fructose-1,6-bisphosphate (F1,6BP), glyceraldehyde-3-phosphate (GAP), and dihydroxyacetone phosphate (DHAP) (Fig 6A–6E). On the other hand, Refed gWAT exhibited enhanced glycolysis throughout the pathway, particularly in the lower glycolysis, indicated by higher levels of 3-phosphoglycerate (3PG), 2-phosphoglycerate (2PG), phosphoenolpyruvate (PEP), and pyruvate (Fig 6F–6I). These results imply that glycolysis-related pathways in BAT might undergo different alterations during refeeding. Supporting this, the glycerol-3-phosphate (G3P) level was elevated (Fig 6J), and the mRNA levels of enzymes linked to G3P production, such as glycerol-3-phosphate dehydrogenase (GPD1) and triosephosphate isomerase (TPI), were induced in Refed BAT (Fig 6L). Additionally, the upper glycolytic enzymes hexokinase (HK2), glyceraldehyde 3-phosphate dehydrogenase (GAPDH), and phosphoglycerate kinase 1 (PGK1) were upregulated in Refed BAT, while pyruvate kinase (PKM) remained unaffected (Fig 6L). Lactate levels were also significantly increased in Refed BAT but showed only a slight rise in gWAT (Fig 6K). These findings suggest that periodic fasting and refeeding robustly activate upper glycolysis and its offshoots, reprogramming glucose metabolism in BAT (Fig 6M). Food intake during refeeding increased by approximately 79.6% compared to the AD group (S1B Fig), which was likely enhancing the uptake of dietary FAs, glucose, and amino acids by BAT.

Consistent with the increased production of G3P in refed BAT, large lipid droplets were observed in both Refed BAT and the liver (S1D Fig). TG synthesis was also induced in Refed BAT, as indicated by the upregulation of enzymes Agpat2 but not Agpat3, and Dgat2, which are involved in the biosynthesis of phospholipids and significant TG formation. This was accompanied by a notable decline in FFAs (Figs 2, 3, and S5A). These findings suggest that the increase in TG content and tissue volume in Refed BAT may be attributed to G3P production via glyceroneogenesis and extensive TG synthesis. Additionally, like findings in other studies [6–14], periodic fasting and refeeding led to the induction of thermogenic markers UCP1 and PGC1α, though not C/EBPβ, supporting metabolic reprogramming in Refed BAT (S5B and S5C Fig). The expression levels of FA transport protein FATP1, also known as SLC27A1, were significantly upregulated by refeeding despite a slight induction on CD36, SCD5, ELOVL6, and ChREBPα (MLXIPL) (S5D and S5E Fig). Supporting this, Refed animals exhibited a gradual increase in energy expenditure and respiratory exchange ratio during the refeeding phase after three cycles of periodic fasting and refeeding, despite a high reliance on FAs during the fasting phase (S5F, S5G, and S6A–S6D Fig). Whereas under thermoneutral conditions, BAT whitening and lipid remodeling were attenuated, although liver remodeling became more pronounced (S7A–S7F Fig; S2 Raw Images; S4 Table), suggesting that the distinct adaptation of BAT corresponds to moderate cold conditions.

In support of the increased upper glycolysis observed in Refed BAT, glycolytic shunts leading to glycogenesis and the PPP were notably activated by ADF, primarily in BAT (S8A–S8I Fig and S1 Table). This stimulation of glycogenesis and the PPP was evidenced by significant elevations in key metabolites, including glucose-1-phosphate, 6-phosphogluconate (6-PG), ribulose-5-phosphate (Ru5G), ribose-5-phosphate (R5G), sedoheptulose-7-phosphate (S7G), and NADPH (S8A–S8F Fig). Additionally, electron microscopy analysis and PAS staining revealed glycogen accumulation in Refed BAT (S8G Fig). Correspondingly, the transcription of enzymes involved in the PPP, such as glucose-6-phosphate dehydrogenase (G6PD) and 6-phosphogluconate dehydrogenase (PDG), was upregulated in Refed BAT. However, the expression levels

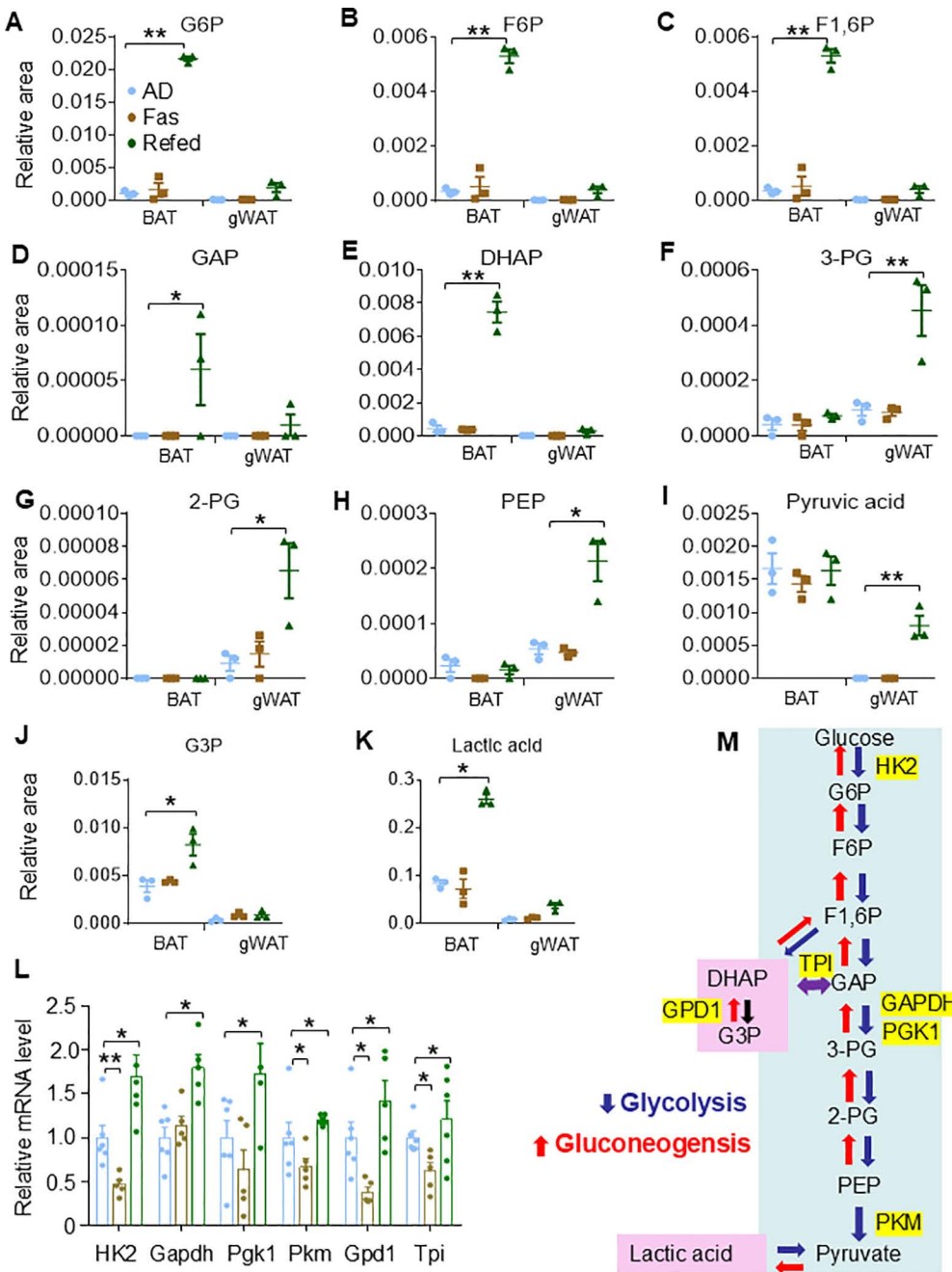

**Fig 6. ADF stimulates upper glycolysis and offshoots of glycolysis-production of G3P and lactate in BAT.** ADF dramatically stimulated the upper glycolysis and off shoots of lower glycolysis in BAT indicated by the levels of glucose 6-phosphate (G6P) **(A)**, fructose 6-phosphate (F6P) **(B)**, fructose 1,6-diphosphate (F1,6P) **(C)**, glyceraldehyde 3-phosphate (GAP) **(D)**, dihydroxyacetone phosphate (DHAP) **(E)**, 3-phosphoglyceric acid (3-PG) **(F)**, 2-phosphoglyceric acid (2-PG) **(G)**, phosphoenolpyruvic acid (PEP) **(H)**, pyruvic acid (pyruvate) **(I)**, glycerol-3-phophate (G3P) **(J)**, and lactate **(K)**. **L.** Refed BAT showed upregulated mRNA levels of enzymes for glycolysis and production of G3P, including hexokinase 2 (HK2), glyceraldehyde 3-phosphate dehydrogenase (GAPDH), phosphoglycerate kinase 1 (PGK1), and glycerol 3-phosphate dehydrogenase 1 (GPD1) in BAT ($n = 6$/group). **M.** The summary of glycolysis pathway highlighting G3P production and enzymes. The raw data for 6L is provided in S1 Table, and the raw data of 6A-6K are presented in S2 Table. The $T$ Test was used to analyze Fig 6. Data are presented as mean ± SEM. *$P < 0.05$, **$P < 0.01$.

of phosphoglucomutase (PGM) and glycogen phosphorylase (GYP) did not show significant changes due to periodic fasting and refeeding in BAT (S8H Fig).

**mTORC1 is required for periodic fasting and refeeding-elicited shift of lipids from high to low degree of unsaturation and increased TG synthesis in BAT**

We then investigated whether insulin signaling and energy sensor mTORC1 are involved in the shift of lipids from high to low unsaturation, as well as TG formation in BAT. We found that periodic fasting and refeeding had no significant effect on the phosphorylation of Akt at Thr308 and Ser473 in BAT (Figs 7A and S9A and S3 Raw Images). However, refeeding markedly activated the phosphorylation of S6K, with a slight decrease observed during fasting compared to the AD group (Figs 7A and S9A and S3 Raw Images), indicating aberrant mTORC1 signaling in Refed BAT. To determine whether mTORC1 plays a role in the metabolic reprogramming induced by IF, we administered AAV-Cre virus into the BAT of 10-week-old mice with floxed Raptor, a positive regulatory subunit of the mTORC1 complex [30]. Two weeks postvirus injection, the mice underwent three cycles of ADF. Western blot analysis confirmed successful Raptor suppression in BAT, though some extent of suppression was observed in the liver, indicating potential viral leakage (Figs 7B and S9B and S3 Raw Images). The suppression of Raptor diminished the IF-induced whitening of BAT and the accumulation of large lipid droplets, with minimal effects on gWAT and inguinal WAT (iWAT) (Fig 7C and 7D). Additionally, the mTORC1 inhibition suppressed the hepatic steatosis associated with IF (Fig 7D), which might be caused by the leaking of virus (Fig 7C). MALDI analysis revealed that the increased lipids, including TGs (50:1 and 50:2), MG (22:3), LPC (18:0), PC (34:0 and 36:2), and HexCer 32:6, by refeeding were diminished by Raptor deficiency in BAT (Fig 7E and 7F and S4 Raw Images). Whereas the decrease in PS (38:8) by refeeding was not significantly affected by mTORC1 inhibition (Fig 7E and 7F). These results suggest that mTORC1 may partially mediate IF-induced lipid reprogramming. Consistent with, refeeding-induced SLC27A1 was attenuated by mTORC1 inhibition in BAT (Figs 7G and S9C and S3 Raw Images).

## Discussion

BAT functions as an effective energy sink that disposes excess lipids, amino acids, and glucose, leading to a profound improvement in energy and glucose homeostasis [18,24,50–53]. In addition, studies in humans and rodents have highlighted the beneficial effects of BAT on cardio-metabolic health [18,54–58]. However, the precise regulation and integration of various metabolic pathways in BAT are still ill-understood. Using an omics approach and spatial tissue imaging, this study aims to elucidate the effects of periodic fasting and refeeding—a promising therapeutic strategy for metabolic disorders—on spatial metabolic landscape. Results reveal that BAT and WAT exhibit distinct lipid profiles, with BAT showing a preference for unsaturated free and lipid-bound FAs. In addition, periodic fasting and refeeding elicits a shift from highly unsaturated to more saturated lipids, leading to spatial and dynamic reprogramming of lipid metabolism in BAT. This study uncovers a novel BAT lipid landscape, highlighting its distinct adaptation to fasting regimen.

Despite extensive studies on lipid metabolism in BAT, particularly lipolysis—a pathway linked to the uncoupling process and heat production—our understanding of the BAT lipid profile remains limited [59]. Advances in omics technology have significantly propelled this field forward. It has been documented that BAT has a higher abundance of phospholipids, such as PEs and PCs, compared to WAT, and that these phospholipids contain longer PUFAs [60,61]. NMR spectroscopy has been used to detect the high level of unsaturation in BAT compared to WAT [62]. These findings not only support the fluidity of BAT membranes [59], but also suggest a potential role for phospholipid dynamics in the thermogenic program of BAT. In support of this, our study showed that VLC-PUFAs (≥ 20 carbons) and 13/14-carbon FAs distinguish murine BAT from WAT, whereas C15-19 FAs are more enriched in WAT (Figs 2-3). These characteristics of both fat depots become more pronounced during fasting but diminish upon refeeding, indicating a reshaped lipid species profile during periodic fasting and refeeding. Schweizer and colleagues also found that longer PUFAs are enriched in brown and beige adipocytes [63]. They identified that elongase Elovl3, an enzyme responsible for the elongation of FAs, plays a critical role in

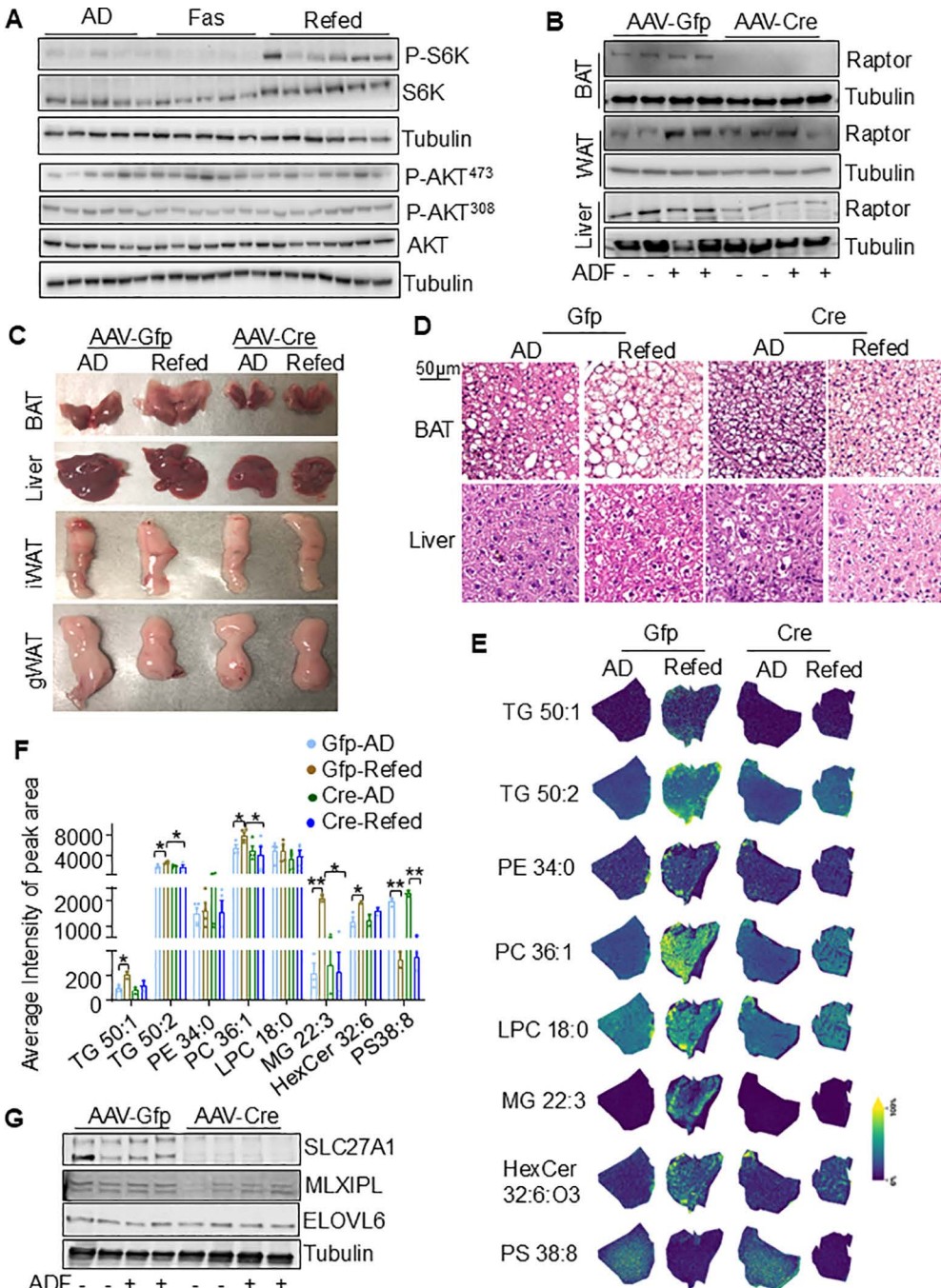

**Fig 7. Periodic fasting and refeeding re-shapes lipid saturation, storage, and distribution via mTORC1. A.** The phosphorylation levels of S6K were notably induced by refeeding in BAT despite a slight increase in AKT phosphorylation ($n = 6$–8/group). **B.** The protein levels of Raptor were decreased by AAV-Cre in Raptor floxed BAT and liver but not in gWAT ($n = 4$–5/group). **C.** Representative images of gWAT, iWAT, BAT, and liver in AD and Refed after injection of AAV-Cre or AAV-GFP ($n = 4$–5/group). **D.** H&E staining of gWAT, iWAT and BAT and Liver after injection of AAV-Cre or AAV-GFP ($n = 4$–5/group). **E.** Specific lipids such as TGs, PCs, and LPC with low degrees of unsaturation, as well as MG and Hex-Cer were elevated upon refeeding, which was reversed by inactivation of mTORC1 in BAT ($n = 4$–5/group). **F.** Quantification of specific lipids in Fig 7E. **G.** Western blots of SLC27A1, MLXIPL, and ELOVL6 in BAT ($n = 4$–5/group). The raw data of western blot in this figure (7A, 7B, and 7G) are presented in S3 Raw Images, and raw data for MALDI analysis 7E is provided in S4 Raw Images. The raw data for 7F is presented in S1 Table. The *T* Test was used to analyze Fig 7E. Data are presented as mean ± SEM. *$P < 0.05$, **$P < 0.01$.

PLOS Biology

the thermogenic program of cells [63]. However, the relationship of VLC-PUFAs and 13/14-carbon FAs in synthesis and whether they are key substrates for the uncoupling process remains to be explored.

In general, adipose tissue contains a vast array of lipid species, including isobaric and isomeric lipids, which can be difficult to distinguish and quantify. Therefore, achieving high spatial resolution to accurately map lipid distributions in adipose tissue has been challenging. In the present study, using MALDI-MSI, we successfully mapped the spatial distribution of lipids in BAT. Consistent with LC- and CE-TOFMS analyses, MSI showed that BAT is enriched with highly unsaturated glycerophospholipids and glycerolipids, including PI-Cer, PS, PC, and TG, which are dynamically altered during periodic fasting and refeeding, ending with a marked decline upon refeeding (Figs 4 and 5). In contrast, relatively saturated glycerolipids surge in Refed BAT. Spatially, these lipids rapidly accumulate in specific areas during refeeding, forming distinct "sparkle regions" with dramatically elevated levels. These dynamic regions correspond to vascular zones, which tend to disappear when transitioning to fasting. These results further support the hypothesis that newly influx fatty acids (FFAs), with relatively high degree of saturation, are rapidly esterified upon cellular enter. These "sparkle regions" of lipids suggest the formation of large lipid droplets to adapt the loss of phospholipids during fasting. The FA composition, rather than the total TG levels, has been linked to aging. TGs composed of saturated or unsaturated FAs that were short (≤C14) or large (≥C24) but not total TG levels are accumulated upon aging [64]. In addition, HFD significantly suppresses the expression of desaturases SCD1/2 and elongase Elovl6 [65] and de novo adipogenesis, increasing the saturation levels of BAT fats.

In this study, we observed that BAT exhibits a distinct adaptation to acute ADF compared to WAT, characterized by a form of "metabolic memory" that facilitates enhanced nutrient uptake during refeeding in response to prior energy deficits. During fasting, WAT releases substantial amounts of FFAs to meet systemic energy demands and subsequently replenishes its lipid stores during refeeding, an expected metabolic response. In contrast, BAT does not appear to contribute significantly to systemic energy supply during fasting. Instead, it likely takes up lipids released from WAT to sustain thermogenesis under moderate cold conditions, as supported by recent studies [21,66]. Consequently, fasting alone does not markedly alter the lipid profile of BAT. However, refeeding induces a pronounced influx of FAs into BAT, which are rapidly esterified into various glycolipids, leading to the expansion of lipid droplets. Although these incoming FAs may be relatively low in saturation, their accumulation results in an overall increase in lipid saturation within BAT, contributing to its whitening. This adaptive response in BAT may serve to buffer the liver from lipid overload, functioning as a form of "metabolic memory" that enhances nutrient uptake during refeeding in response to prior energy deficits. In line with this, BAT preferentially utilizes glucose over FAs for thermogenesis during the transition from fasting to refeeding, following acute periodic fasting and refeeding.

As a well-known energy sensor, mTORC1 acts as a master regulator of intracellular metabolism, including lipid metabolism involving de novo lipogenesis and lipolysis [32,35–38]. mTORC1 is essential for the development of various fat depots, such as brown and white fat tissues [43–45]. Beyond development, mTORC1 also regulates thermogenesis, a process dependent on lipolysis [35,36,39–45]. However, the regulatory effects of mTORC1 on thermogenic programs remain controversial, with ongoing debate about whether mTORC1 facilitates or inhibits PKA activation [35,36,39,40]. The sustained thermogenic program in BAT necessitates rapid lipid turnover, which facilitates metabolic adaptation to environmental fluctuations. In this study, we observed spatial lipid turnover in brown fat during periodic fasting and refeeding. Despite low baseline levels, more saturated LPCs, PEs, PCs, and TGs significantly increased in Refed BAT, accompanied by enhanced angiogenesis, forming "sparkle regions." Conversely, lipids with a high degree of unsaturation exhibited high baseline levels but underwent spatial redistribution during fasting, followed by a notable decrease in Refed BAT. Inhibiting mTORC1 in BAT blocked IF-elicited lipid turnover and the shift from high to low unsaturation for a variety of lipids (Fig 7). These results highlight the spatial and dynamic distribution of lipids in BAT, which may be critical for thermogenic adaptation to dietary stress. Additionally, Refeeding promotes intracellular FA transport by inducing the expression of SLC27A1, an effect that is attenuated when mTORC1 is inhibited in BAT, suggesting that mTORC1 partially mediates the lipid remodeling response elicited by refeeding.

The metabolomic and lipidomic analyses provided evidence that thiamine metabolism may be correlated with the reprogramming of carbohydrate metabolism in Refed BAT. Repeated fasting and refeeding suppress thiamine transporter 2 (ThTr2) expression and decrease the levels of thiamine and its metabolites, thiamine pyrophosphate (TPP) and thiamine monophosphate (TMP), in BAT (S10A–S10F Fig). As a cofactor of pyruvate dehydrogenase (PDH) [67–69], TPP plays a critical role in the breakdown of sugars and protein building blocks (amino acids) [70,71]. The decrease in TPP due to refeeding reduces the conversion of pyruvate to acetyl-CoA and the substantial TCA cycle (S11A–S11I Fig). The decrease in TPP in Refed BAT at least partially explains the selective upregulation of upper glycolysis, glycolytic shunts, and glycolytic offshoots, leading to the production of G3P and lactate. Along with this, BAT glyceroneogenesis is markedly stimulated by refeeding, considering the much higher levels of lactic acid, fumaric acid, malic acid, DHAP, and G3P during ADF. As a result, the levels of other glycolysis intermediate metabolites (3-PG, 2-PG, PEP, and pyruvic acid) in BAT were similar among the three groups, which may be explained by the neutralization of glyceroneogenesis. In addition to glyceroneogenesis, G-3-P can also be formed by the phosphorylation of glycerol via glycerol kinase or by the reduction of dihydroxyacetone phosphate via G-3-P dehydrogenase. Glycerol kinase, although highly active in the liver, is present at low activity in adipose tissue and skeletal muscle [72–74]. Therefore, glucose metabolism pathways, including glyceroneogenesis, glycolysis, and glycolytic shunts, are highly sensitive to periodic fasting and refeeding.

In summary, the present study demonstrated that BAT has a distinct lipid profile compared to WAT, favoring VLC-PUFAs and C13-14 FFAs and highly unsaturated glycerophospholipids, glycerolipids, and sphingolipids, showing distinct lipid distribution dynamics. Periodic fasting and refeeding lead to a spatial redistribution, stimulating the accumulation of unsaturated fats during fasting and more saturated fats upon refeeding. This shifts the lipids from high to low unsaturation in BAT during the refeeding phase, which is partially mediated by mTORC1 signaling. These results highlight a unique BAT adaptive mechanism in response to periodic fasting and refeeding, characterized by a surge in FA influx, elevated glycolipid synthesis, and subsequently increased saturation of tissue lipids.

### Limitations of the present study

In the present study, only male mice were characterized. In addition to BAT, the virus-mediated inactivation of mTORC1 also affects hepatic signaling, suggesting a potential role of liver in BAT lipid remodeling. Supporting this, the increase in liver mass during refeeding becomes more pronounced when lipid remodeling in Refed BAT is attenuated under thermoneutral conditions (S7 Fig and S2 Raw Images). The observed correlation between decreased thiamine levels and lipid remodeling points to a possible role for thiamine in regulating glycerol production, glyceroneogenesis, and TG accumulation. However, whether thiamine metabolism serves as a regulatory checkpoint in lipid remodeling remains unclear. Tracer experiments could offer valuable insights into the potential link between thiamine metabolism and lipid remodeling and warrant further investigation.

### Supporting information

**S1 Fig. Periodic fasting and refeeding rapidly stimulates BAT whitening. A.** Schematic diagram of Alternate Day Fasting (ADF). AD, ad libitum; Fas, ADF ended with fasting; Refed, ADF ended with refeeding. **B.** The food intake was notably decreased upon refeeding following with 24-h fasting. **C.** Body weight significantly decreased after fasting and was regained after refeeding. **D.** The representative images of H&E staining for gWAT, iWAT and BAT and Liver. The raw data for S1B and S1C are presented in S1 Table. Data in S1B and S1C Fig are presented as mean ±SEM. *T* Test was used to analyze the data. *$P < 0.05$, **$P < 0.01$.
(TIF)

**S2 Fig. Microbial metabolites and dipeptides are upregulated by perioidic fasting and refeeding within gWAT.** The basal levels of microbial metabolites **(A)** and dipeptides **(B)** were significantly higher in gWAT compared to BAT and were

further increased by refeeding compared to the AD and Fas groups. The raw data of metabolomics and lipidomics for S2A and S2B are presented in S2 Table.
(TIF)

**S3 Fig. Unsaturated fatty acids are highly enriched in BAT compared to gWAT; however, their levels decrease upon refeeding.** BAT displayed higher basal levels of cholesterol **(A)**, 7-dehydrocholesterol-1 **(B)**, and 7-dehydrocholesterol-3 **(C)**, but not 22-hydroxycholesterol **(D)** compared with gWAT. Moreover, the levels of bioactive lipids including anandamide **(E)**, oleoyl ethanolamide **(F)**, palmitolethanolamide **(G)**, and stearoyl ethanolamide **(H)** were higher in BAT than those in gWAT under both AD and fasting conditions. **I.** Alignment of representative lipid (LPC16:0) images (A, D) with H&E staining (B, C) in a representative BAT section. **J.** The enriched LPC16:0 "sparkles" were correlated with CD31 expression. The raw data of metabolomics and lipidomics for S3A–S3J are presented in S2 Table. S3A–S3H Fig was analyzed by *T* Test. The data are presented as mean±SEM. *$P<0.05$, **$P<0.01$.
(TIF)

**S4 Fig. The immunofluorescence staining of UCP1 and CD31 in BAT. A**. The enriched LPC16:0 "sparkles" were not correlated with UCP1-high adipocytes. **B**. Representative H&E staining images of all BAT tissue samples. **C**. Representative CD31 staining images of all BAT tissue samples. **D**. Representative UCP1 staining images of all BAT tissue samples.
(TIF)s

**S5 Fig. Acute periodic fasting and refeeding activates the genes involved in TG synthesis and thermogenesis in BAT.** The mRNA expression of genes encoding the enzymes for TG synthesis, De novo lipogenesis, and Lipolysis pathway during periodic fasting and refeeding. **B, C**. The protein level of thermogenesis markers in BAT during periodic fasting and refeeding. **D.** The western blot of fatty acids desaturases, elongase, and transport genes. **E.** Quantification of protein levels from the Western blots shown in S4D Fig. **F, G.** Oxygen consumption during ADF for 3 cycles, using the CaIR website as described in the methods. The raw data for S5A, S5C, S5E, and S5G are presented in S1 Table. The uncropped western blot images for panel S5B and S5D are provided in S3 Raw Images. Data are presented as the mean±SEM. *$P<0.05$; **$P<0.01$.
(TIF)

**S6 Fig. The Respiratory Exchange Ratio and activity in response to Acute periodic fasting and refeeding. A, B.** ADF mice exhibited significantly lower Respiratory Exchange Ratio (RER) during fasting phase but gradually higher RER during refeeding than ad libitum controls. **C, D.** Refed mice showed a slight increase in motor activities. The raw data for S6B and S6D are presented in S1 Table. Data are presented as the mean±SEM. *$P<0.05$; **$P<0.01$.
(TIF)

**S7 Fig. Refeeding-induced BAT whitening and lipid remodeling were diminished under thermoneutral conditions. A**. The Representative images of gWAT, iWAT, BAT, and liver in three groups ad libitum (AD), fasting of ADF (Fas), and refeeding of ADF (Fed) under thermoneutral conditions (30 °C). **B**. The mass of various fat tissues and liver after periodic fasting and refeeding. The food intake **(C)** and body weight **(D)** during alternative-day fasting under thermoneutral conditions. **E.** H&E staining of fat depots. **F.** The MSI showed a slight decrease for PC 36:4, PC 36:6, PC 38:4, and PE 44:4 in BAT upon refeeding. The raw data for S5B–S5D are provided in S1 Table, and the raw data of MALDI for S7F is presented in S2 Raw Images. Data are presented as the mean±SEM. *$P<0.05$; **$P<0.01$.
(TIF)

**S8 Fig. Periodic fasting and refeeding rapidly stimulates glycogenesis, PPP, and glyconeogenesis in BAT.** The alterations of intermediate metabolites within glycogenesis and PPP pathways in BAT and gWAT during periodic fasting and refeeding; refeeding activated glycogenesis and PPP pathways in BAT as indicated by Glucose 1-Phosphate (G1P)

(**A**), 6-Phosphogluconic acid (6-PG) (**B**), Ribulose 5-Phosphate (Ru5P) (**C**), Ribose 5-Phosphate (R5P) (**D**), Sedoheptulose 7-Phosphate (S7P) (**E**), and Nicotinamide Adenine Dinucleotide Phosphate (NADPH) (**F**), albert to a lesser extent in gWAT. **G**. Feast BAT exhibited significant glycogen accumulation indicated by TEM and PAS staining. **H**. The mRNA levels of the enzymes involved in glycogenesis and PPP, including Glycogen Phosphorylase (GYP), Glucose-6-Phosphate Dehydrogenase (G6PD), and 6-Phosphogluconate Dehydrogenase (PGD) but not Phosphoglucomutase (PGM), were induced by refeeding in BAT. **I**. The summary of glyconeogenesis and PPP pathways. The raw data for S8H is presented in S1 Table, and the raw data for metabolomics and lipidomic (S8A–S8F) are provided in S2 Table. All data in this Figure were analyzed by *T* Test and are presented as mean±SEM. *$P<0.05$, **$P<0.01$.
(TIF)

**S9 Fig. The effects of Raptor knockdown on refeeding-induced lipid remodeling in BAT. A**. Quantification of phosphorylation protein levels from the Western blots shown in Fig 7A. **B**. Quantification of Raptor protein levels from the Western blots shown in Fig 7B. **C**. Quantification of protein levels from the Western blots shown in Fig 7G. The raw data for S9A–S9C are presented S1 Table. All data in this Figure were analyzed by *T* Test and are presented as mean±SEM. *$P<0.05$, **$P<0.01$.
(TIF)

**S10 Fig. Periodic fasting and refeeding selectively suppresses thiamine transportation and metabolism in BAT.** The intermediate metabolites within thiamine metabolism were altered during periodic fasting and refeeding. Refeeding notably reduced thiamine metabolites including thiamine (**A**), thiamine phosphate (**B**), and thiamine diphosphate (**C**) in BAT despite a slight increase in thiamine of gWAT. **D.** The mRNA levels of thiamine transporters and enzymes were highly expressed in BAT compared to gWAT. **E.** The mRNA levels of thiamine transporters and enzymes involved in thiamine metabolism in BAT were differentially altered during periodic fasting and refeeding. **F.** The summary of the glycolysis and thiamine pathways. The levels of Acetyl-CoA (AcCoA). The raw data for S10D and S10E are presented in S1 Table, and all metabolomics and lipidomic data for S10A–S10C are provided in S2 Table. Data in S10D and S10E Fig are presented as the mean±SEM. *$P<0.05$; **$P<0.01$.
(TIF)

**S11 Fig. Periodic fasting and refeeding reduces the conversion of pyruvate to acetyl-CoA and the substantial TCA cycle.** (**A**), Citric acid (**B**), Fumaric acid (**C**), Malic acid (**D**), Alanine (Ala) (**E**), Glutamine (Gln) (**F**), and NAD+ (**G**) in BAT**. H.** Expression levels of mRNA of glyceroneogenesis pathway enzymes increased by refeeding of ADF in BAT. **I.** The summary of pyruvate destinations. The raw data for S11H is presented in S1 Table, and all metabolomics and lipidomic data for S11A–S11G are provided in S2 Table. Data in S11A–S11H are presented as the mean±SEM. *$P<0.05$; **$P<0.01$.
(TIF)

**S1 Table. Individual numerical values corresponding to Figs 1–7 and S1–S11 Figs are provided.** The values are organized by figure in separate sheets (one sheet per figure).
(XLS)

**S2 Table. The metabolomic and lipidomic datasets.** B-AD, B-Refed, and B-Fas are referred to brown adipose tissue under different conditions, ad libitum (AD) and alternate day fasting ended with refeeding (B-Refed) or fasting (B-Fas), respectively; e-AD, e-Refed, and e-Fas are referred to epididymal white adipose tissue under different conditions, ad libitum (AD) and alternate day fasting ended with refeeding (e-Refed) or fasting (e-Fas) (*n*=3/group), respectively. For Fig 1D, PCA score, PC1, and PC2 show the first principal component and second principal component, respectively. The number in parentheses is the contribution rate, and the plot labels are sample names. Principal Component Analysis (PCA) was performed using statistical analysis software (developed at HMT). B-AD, B-Refed, and B-Fas refer ad libitum

(AD) and alternate day fasting ended with refeeding (B-Refed) or fasting (B-Fas) in brown adipose tissue; e-AD, e-Refed, and e-Fas refer ad libitum (AD) and alternate day fasting ended with refeeding (e-Refed) or fasting (e-Fas) in white adipose tissue ($n=3$/group). For Putative Metabolites, Peak ID consists of analysis mode and number. The alphabets indicate measurement mode; Cation (C) and Anion (A) mode. Putative metabolites listed in "Compound name" were assigned based on $m/z$ and MT. Those listed in "PubChem ID/ HMDB ID/ peptide" were assigned based on $m/z$ only. "N.D." and "N.A" represent "Not Detected" and "Not Available", respectively. "Ratio" was calculated between two indicated groups (left: numerator, right: dominator). "$p$-value" was calculated using $t$ test. B-AD, B-Refed, and B-Fas refer ad libitum (AD) and alternate day fasting ended with refeeding (B-Refed) or fasting (B-Fas) in brown adipose tissue; e-AD, e-Refed, and e-Fas refer ad libitum (AD) and alternate day fasting ended with refeeding (e-Refed) or fasting (e-Fas) in white adipose tissue ($n=3$/group).
(XLSX)

**S3 Table. Average intensity (Peak Area) analysis in BAT under room temperature (22 ℃) ($n=3$/group).** Statistical analysis of the rest of data was performed using a two-tailed Student $t$ test between two groups.
(XLSX)

**S4 Table. The representative MALDI lipids average intensity (Peak Area) analysis in BAT under thermoneutrality condition (30 ℃) ($n=3$/group).** Statistical analysis of the rest of data was performed using a two-tailed Student $T$ test between two groups.
(XLSX)

**S1 Raw Images. The raw image of MALDI-MSI in Figs 4 and 5.** The results showed lipid and metabolite profile of BAT under AD, Refed, and FAS conditions at room temperature (22 ℃) ($n=3$/group).
(PDF)

**S2 Raw Images. The raw image of MALDI-MSI in S7F Fig. The results showed the different lipid and metabolite profile of BAT under AD, FAS, and Refed conditions at thermoneutral condition (30 ℃) ($n=3$/group).**
(PDF)

**S3 Raw Images. The uncropped western blot images for Figs 7A, 7B, 7G, S5B, and S5D.**
(PDF)

**S4 Raw Images. The raw image of MALDI-MSI in Fig 7E and 7F.** The results showed lipid and metabolite profile in Raptor KO BAT under AD, Refed, and FAS conditions at room temperature (22 ℃) ($n=5$–6/group).
(PDF)

## Acknowledgments

We thank the Metals in Biology and Disease Center for the MALDI-TOF analysis and Autophagy, Inflammation and Metabolism Center at UNMHSC for providing the Chemidoc, for our present study. We also want to thank the Department of Biochemistry and Molecular Biology for the EVOS Imaging System and the Department of Cell Biology and Molecular Biology for the Cryogenic Electron Microscopy at the UNMHSC, for our present study.

## Author contributions

**Conceptualization:** Changjian Feng, Meilian Liu.

**Data curation:** Xing Zhang, Ting Jiang, Chunqing Wang, Valeria F. Montenegro Vazquez, Dandan Wu, Xin Yang, Que Le, Melody S. Sun, Xiaofei Wang, Meilian Liu.

**Formal analysis:** Xing Zhang, Ting Jiang, Chunqing Wang, Valeria F. Montenegro Vazquez, Dandan Wu, Xin Yang, Que Le, Melody S. Sun, Xiaofei Wang, Jing Pu, Meilian Liu.

**Funding acquisition:** Xing Zhang, Matthew Campen, Changjian Feng, Meilian Liu.

**Investigation:** Xing Zhang, Ting Jiang, Meilian Liu.

**Methodology:** Xing Zhang, Ting Jiang, Chunqing Wang, Dandan Wu, Xin Yang, Xuexian O. Yang, Changjian Feng.

**Project administration:** Xing Zhang, Changjian Feng, Meilian Liu.

**Resources:** Xing Zhang, Xuexian O. Yang, Matthew Campen, Changjian Feng, Meilian Liu.

**Software:** Ting Jiang, Changjian Feng.

**Supervision:** Xuexian O. Yang, Jing Pu, Matthew Campen, Changjian Feng, Meilian Liu.

**Validation:** Xing Zhang, Ting Jiang, Chunqing Wang, Dandan Wu, Xin Yang, Que Le, Melody S. Sun, Xiaofei Wang, Meilian Liu.

**Writing – original draft:** Xing Zhang, Meilian Liu.

**Writing – review & editing:** Xing Zhang, Valeria F. Montenegro Vazquez, Changjian Feng, Meilian Liu.

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
