## [Editor Report · Decision Letter 0]

15 Apr 2025

Dear Dr Liu,

Thank you for submitting your manuscript entitled "Periodic Fasting and Refeeding Re-shapes Lipid Unsaturation Degree and Spatial Distribution in Brown Adipose Tissue" for consideration as a Research Article by PLOS Biology and I am sorry for the delay in sending you an initial decision. We had wished to discuss your paper with an Academic Editor, with relevant expertise, but unfortunately were not able to secure feedback over the last week.

In the absence of input from an expert in the field, I have discussed your paper with my colleagues on the editorial team, and I am writing to let you know that we would like to send your submission out for external peer review. While we are, in principle, interested in the study, I should note that at this stage we did not feel able to make a firm call about whether the study offers the level of advance that we aim to publish and so we will be looking for strong support from the reviewers in that regard to move forward with the study after review. As a last note, we think your paper is currently best suited for our 'Methods and Resources' article type, and so are planning on reviewing it as such. We think this will have the highest chance of a positive outcome.

**Before we can send your manuscript to reviewers, we need you to complete your submission by providing the metadata that is required for full assessment. To this end, please login to Editorial Manager where you will find the paper in the 'Submissions Needing Revisions' folder on your homepage. Please click 'Revise Submission' from the Action Links and complete all additional questions in the submission questionnaire.

Once your full submission is complete, your paper will undergo a series of checks in preparation for peer review. After your manuscript has passed the checks it will be sent out for review. To provide the metadata for your submission, please Login to Editorial Manager (https://www.editorialmanager.com/pbiology) within two working days, i.e. by Apr 17 2025 11:59PM.

Kind regards,

Luke

Lucas Smith, Ph.D.

Senior Editor

PLOS Biology

lsmith@plos.org

---

## [Decision Letter · Decision Letter 1]

12 Jun 2025

Dear Meilian,

Thank you for your patience while your manuscript "Periodic Fasting and Refeeding Re-shapes Lipid Unsaturation Degree and Spatial Distribution in Brown Adipose Tissue" was peer-reviewed at PLOS Biology. Your manuscript has been evaluated by the PLOS Biology editors, an Academic Editor with relevant expertise, and by several independent reviewers.

As you will see in the reviewer reports, which can be found at the end of this email, although the reviewers find the work potentially interesting, they have also raised a substantial number of important concerns. Based on their specific comments and following discussion with the Academic Editor, it is clear that a substantial amount of work would be required to meet the criteria for publication in PLOS Biology. However, given our and the reviewer interest in your study, we would like to invite a comprehensive revision of the study that thoroughly addresses the reviewers' comments. Given the extent of revision that would be needed, we cannot make a decision about publication until we have seen the revised manuscript and your response to the reviewers' comments. Your revised manuscript would need to be seen by the reviewers again, but please note that we would not engage them unless their main concerns have been addressed.

We had initially been considering your paper as a "Resource" article, but in light of the reviewer feedback we think you will need to develop the study further to better support the conclusions and show the physiological relevance of the findings - and that the revision would likely be better suited for our normal Research Article format. We think the revision will need respond to all reviewer comments point-by-point, and will particularly need to provide (1) validation the mTORC1 regulation of lipid phenotypes; (2) an assessment of the biological importance of lipid remodeling; (3) validation of key findings under thermoneutral conditions; and (4) supporting western blot and IHC data.

We would not require you to provide inter-organ tracing and sc/spatial transcriptomic experiments in the revision, as proposed by reviewer 3. We think these experiments would be interesting, but that those points could be addressed through discussion of limitations.

We appreciate that these requests represent a great deal of extra work, and we are willing to relax our standard revision time to allow you 6 months to revise your study. Please email us (plosbiology@plos.org) if you have any questions or concerns, or envision needing a (short) extension.

**IMPORTANT - SUBMITTING YOUR REVISION**

*Resubmission Checklist*

*Published Peer Review*

*PLOS Data Policy*

*Blot and Gel Data Policy*

Sincerely,

Luke

Lucas Smith, Ph.D.

Senior Editor

PLOS Biology

lsmith@plos.org

REVIEWS:

Reviewer #1: The manuscript by Zhang et al details the comprehensive metabolomic analysis of the response to ADF in the mouse model system. This study incorporates innovative techniques such as MALDI-MSI to examine lipid abundance and localisation during fasting/feeding. In addition, they have used an AAV system to examine the effect of mTORC1 disruption in BAT on metabolic pathways. This study contains a wealth of information but has areas that are poorly written and makes some conclusions not convincingly supported by the data presented. Additional experiments and analysis are warranted before the manuscript is ready for publication.

Major comments

**** While the link between mTORC1 and the lipid phenotypes has been suggested by the AAV experiment, there are a number of confounding factors that complicate its interpretation such as the viral leakage and liver effects. The authors should use a different methodology or in vitro system to provide data that supports their conclusions.

**** The authors state in the abstract "Mechanistically, periodic fasting and refeeding activated mTORC1, and genetic inactivation of mTORC1 in BAT diminished ADF induced lipid saturation, storage, and redistribution". I'm not convinced that changes in lipid saturation have been demonstrated by the data shown. Additional experiments/data are required to confirm this point.

**** The data in every panel of Figure 7 needs to be quantified and plots generated with stats testing. The phospho western blots need to be normalised relative to each total protein blot. The raptor westerns normalised to tubulin. The MALDI-MSI data also needs to be quantified, plots drawn and stats presented.

**** BAT can be difficult to dissect from the WAT that lies adjacent to it. How did the authors remove contaminating WAT from the BAT depot?

**** Why did the authors choose this particular ADF intervention length of only 6 days? Other studies use 2 or more weeks of ADF intervention.

**** The authors say that the "activity" of certain pathways have increased or decreased throughout the manuscript, but they are only providing static measurement at a single point in time, please restate this as altered abundance instead.

**** The authors propose that the mRNA measurement in Figure 6L is an accurate surrogate for the measurement of the enzymes themselves, but I disagree. The authors should measure the protein levels of these enzymes using westerns/proteomics. A proteomic analysis of these samples would also add significant insight into how each metabolic pathway is being regulated.

**** The manuscript lacks consistency and clarity throughout, for example, referring to "feast BAT" at many points, is that the refed group or the AD group? Please rewrite this throughout in a consistent manner.

**** The manuscript has issues with grammar and style throughout: "became shrank"

Minor comments

**** Why did this study only analyse male mice?

**** Please add an additional tab in Supplementary Table 1 that contains a detailed legend describing all abbreviations used in the table for animal tissues/groups. Also add details of the statistical tests performed, any p-value adjustment, and the number of animals per group in stats tests.

**** Please provide results of statistical tests for each row in Supplementary Table 2. Also, please add an additional tab that contains a detailed legend describing all abbreviations used in the table for animal tissues/groups. Also add details of the statistical tests performed, any p-value adjustment, and the number of animals per group in stats tests.

**** Colour legends need to be placed on all supplementary figures/plots as per figure 1b

**** The following sentence doesn't make sense: "When lipolytic enzymes are deficient in brown adipocytes, the utilization of fatty acids in circulation is induced and substitutes for BAT TG to sustain cold-induced thermogenesis". What is "the utilization of fatty acids in circulation"?

Reviewer #2: This is a well-executed and ambitious study that integrates metabolomics, lipidomics, and spatial imaging to define how periodic fasting and refeeding remodel lipid composition and distribution in BAT. The authors demonstrate that BAT favors VLC-PUFAs and 13-14 carbon FAs at baseline, which are mobilized during fasting, while refeeding promotes a shift toward more saturated lipids. They further implicate mTORC1 signaling in mediating these lipid changes. Overall, this study presents a technically rich and timely investigation into BAT lipid dynamics. The scope of the analytical platforms employed is impressive, and the spatial lipidomic profiling using MALDI-MSI adds important new dimensions to our understanding of BAT lipid plasticity. The key findings in general are intriguing and well supported by lipidomic data. However, several mechanistic and technical issues limit the conclusions, particularly around the functional relevance of the lipid shift and the specificity of mTORC1 signaling. Additional experiments may be needed to move beyond correlative findings.

Major Points:

1. While the study describes shifts in lipid saturation and composition during fasting/refeeding, it does not establish whether these changes influence thermogenesis, energy expenditure, or systemic metabolism. Are these compositional changes functionally adaptive, or merely reflective of nutrient flux? No data are provided on oxygen consumption, thermogenic response to cold or β3-adrenergic stimulation. These assessments may be essential to validate the biological importance of the observed lipid remodeling.

2. The manuscript posits that lipid saturation increases upon refeeding and contributes to BAT whitening and TG accumulation. However, whether this shift causal for the phenotypes is not tested. Could a pharmacological or genetic manipulation that blocks desaturation (e.g., SCD1 inhibition or Elovl3 knockdown) prevent the whitening or suppress TG accumulation in refeeding conditions? Conversely, does increasing PUFA availability protect against whitening?

3. The MALDI-MSI results show spatial "sparkle" zones enriched in saturated lipids during refeeding. However, these regions are not linked to any known BAT microanatomy. Are they aligned with blood vessels, multilocular zones, or regions of sympathetic innervation? Some validation using co-staining or marker overlays (e.g., UCP1, CD31, perilipin) may help to determine whether these lipid hotspots correspond to functionally distinct areas.

4. The authors observe reduced thiamine levels in BAT upon refeeding and speculate that this limits pyruvate oxidation via PDH, redirecting carbon flux to glyceroneogenesis. However, this idea is not directly tested. Are acetyl-CoA or citrate levels affected? Does thiamine repletion restore PDH flux or attenuate TG accumulation? Including even one tracer experiment would substantially strengthen this part of the model.

Minor Points

* Fig. 1D: PCA results are somewhat difficult to interpret without access to the loading plots. Which metabolites or lipid species drive the separation among conditions?

* Fig. 2-3: Please clarify if these are FFAs or FAs derived from all lipid classes?

* Fig. 4: Are MSI ion intensities normalized to internal standards?

* Fig. 6M: The pathway schematic is useful, but the connection to mTORC1 or the proposed glyceroneogenesis mechanism could be clearer.

* Methods: Please include more technical details for MALDI-MSI acquisition (e.g., laser energy, scan settings, number of pixels/ROI). These are essential for reproducibility.

Reviewer #3: While cold-induced lipolysis and fatty acid oxidation in brown adipose tissue (BAT) are well characterized, the metabolic adaptations of BAT to fasting and refeeding remain incompletely understood. This study provides important insights into how BAT dynamically responds to intermittent fasting (IF), particularly under alternate-day fasting regimens, by undergoing spatially coordinated lipid remodeling. The authors demonstrate that fasting and subsequent refeeding shift the BAT lipid composition from highly unsaturated to more saturated species—a process that appears to be partially mediated by mTORC1 signaling. This is a very interesting and timely study, and the manuscript is generally well written. However, several critical points should be addressed to strengthen the mechanistic conclusions and overall impact prior to publication.

Suggestions/comments:

One of the key physiological changes during fasting is a reduction in core body temperature. Therefore, while the observed lipid remodeling in BAT may be driven by nutritional cues, it is also possible that these changes are secondary to fasting-induced hypothermia. To disentangle the effects of nutrient status from those of body temperature, it would be important to reproduce the observed lipid phenotypes under thermoneutral conditions. Demonstrating that these effects persist independent of cold-induced thermogenesis would significantly strengthen the conclusions of the manuscript.

The authors show that fasting and refeeding reshape the BAT lipidome by reducing polyunsaturated lipid species and enriching saturated lipids, with significant spatial heterogeneity and dynamic redistribution. The observed shift from polyunsaturated to saturated/monounsaturated lipids in specific BAT regions is intriguing. However, it is unclear whether these changes result from de novo lipogenesis and fatty acid elongation/desaturation within BAT or from the uptake of circulating lipids. The authors should consider including expression data for key enzymes and transporters such as SCD1, Elovl6, and fatty acid transporters (e.g., CD36, FATP) to help clarify the underlying mechanism.

Also, the functional consequences of the observed lipid shift are not addressed. What are the potential implications of increased lipid saturation in BAT thermogenesis or metabolic flexibility?

Could the authors speculate on the enzymatic or hormonal signals driving this selective remodeling upon refeeding? Are insulin signaling or nutrient-sensing pathways (e.g., ChREBP, SREBP1c) potentially involved?

The authors report that refeeding selectively activates upper glycolysis, glyceroneogenesis, triglyceride synthesis, and glycolytic shunts in BAT—indicating a distinct metabolic reprogramming from white adipose tissue (WAT) that supports both lipid storage and thermogenic readiness. However, it remains unclear why lower glycolysis is not equivalently upregulated. Does this reflect a bottleneck in glycolytic flux, or is it due to a redirection of intermediates toward anabolic processes like triglyceride synthesis? Additional metabolite profiling or enzymatic activity assays may clarify this point.

The authors demonstrate that mTORC1 activation in BAT is essential for refeeding-induced metabolic reprogramming, including increased triglyceride synthesis and a shift to less unsaturated lipids. In this regard, 1) what were the rationales that make the author assume the involvement of mTORC1 in this shift? 2) also, the mechanistic role of mTORC1 in regulating lipid saturation remains insufficiently explored. 3) How does mTORC1 control key enzymatic steps in lipid remodeling, such as elongation and desaturation? Expression or activity data for downstream effectors (e.g., Elovl family members, SCD1) would help clarify the pathway.

Also, the potential viral leakage into the liver complicates the interpretation of BAT-specific mTORC1 effects. Ex vivo or in vitro studies using isolated brown adipocytes or BAT explants could provide more definitive insight into tissue-autonomous mechanisms.

Could the refeeding-induced lipid and fatty acid changes in BAT originate from other tissues, such as the liver? Specifically, are the unsaturated fatty acids enriched during refeeding synthesized de novo in the liver or mobilized from white adipose tissue (WAT)? How can the authors distinguish whether the observed phenotypes are intrinsic to BAT or secondarily influenced by systemic lipid flux from other organs? Demonstrating tissue-autonomous effects will be essential to strengthen the conclusions.

Have the authors considered using stable isotope tracing (e.g., 13C- or 2H-labeled fatty acids or glycerol) to directly track the source of lipids incorporated into BAT during refeeding? Such experiments could help distinguish contributions from liver-derived triglyceride-rich lipoproteins versus WAT-released free fatty acids, providing mechanistic insight into inter-organ metabolic crosstalk. Perhaps this is not necessary for the current study, it would strengthen the manuscript.

Given the spatial heterogeneity observed in lipid remodeling, is there evidence that specific cell populations within BAT (e.g., mature brown adipocytes, progenitors, or immune cells) differentially contribute to these metabolic adaptations? If needed, the use of single-cell or spatial transcriptomics could clarify cell-type-specific metabolic responses and regulatory networks during fasting-refeeding cycles.

How robust and reproducible are the observed spatial lipid changes in BAT? Could technical factors such as tissue sampling orientation, sectioning depth, or inherent depot heterogeneity account for the spatial differences? Additional details on how spatial resolution was standardized and validated would help assess the reliability of the imaging-based lipidomic analysis.

---

## [Decision Letter · Decision Letter 2]

4 Dec 2025

Dear Meilian,

Thank you for your patience while we considered your revised manuscript "Periodic Fasting and Refeeding Re-shapes Lipid Unsaturation Degree and Spatial Distribution in Brown Adipose Tissue" for publication as a Research Article at PLOS Biology. This revised version of your manuscript has been evaluated by the PLOS Biology editors, the Academic Editor, and reviewer 2.

Based on the reviews and our Academic Editor's assessment of your revision we are likely to accept your study . However, before we can do so, we have a number of editorial requests, which we need you to address in a revision that we think will not take very long. These are detailed below.

**IMPORTANT - Please address the following editorial requests.

1) RESPONSE TO REVIEWERS: You will see that reviewer 2 is fully satisfied by the revision. The Academic Editor has also indicated that you have responded appropriately to the previous reviewer comments. However, we did notice one remaining issue which should be addressed before publication. Specifically, in the last round of review R1 specifically requested quantification for every panel Figure 7 - but we do not see the western blot data quantified anywhere. Please provide the required quantification and include it either in the Supplementary Figures or in the main Figure 7.

2) SUBMISSION FILES: I noticed that your submission includes the most recent version of your manuscript and figures, but also previous versions of these files from the round of review. Please update this, by removing any outdated figures or files, and only including the most recent, revised version of your manuscript, figures, and supplemental information. For future submissions, when submitting the revision, it is best practice to remove the outdated files, as this will prevent confusion (we have, and make available to reviewers, the previous version of your manuscript, even if you remove those files).

3) SUPPLEMENTAL FIGURES: I also had trouble finding your supplemental figures in the most recent submission. Please make sure to include these again in your next submission (or point me to them if they were already included) as we will need to check them over before publication.

4) TITLE: We would like to propose a tweak to your title, which we think will improve its flow. If you agree, we suggest you change your title to:

"Periodic fasting and refeeding re-shapes lipid saturation, storage and distribution in brown adipose tissue"

5) ABSTRACT: Please note that per journal policy, the model system/species studied should be clearly stated in the abstract of your manuscript.

6) ETHICS STATEMENT: Please update your ethics statement, in your methods section, to include the approval number for your animal care and use protocol, approved by the IACUC of New Mexico Health Sciences Center. Please also include the specific national or international regulations/guidelines to which your animal care and use protocol adhered. Please note that institutional or accreditation organization guidelines (such as AAALAC) do not meet this requirement.

7) DATA: You may be aware of the PLOS Data Policy, which requires that all data be made available without restriction: http://journals.plos.org/plosbiology/s/data-availability.

>>Under this policy, we would require that you please deposit and make available all raw metabolomic, lipidomic, and MALDI MSI data on a publicly available repository. Please also update your Data Statement in the submission system to accurately describe where your data can be found.

8) DATA: Note, in addition to the raw data for the abovementioned experiments, we require the underlying data for all other figures presented in your study. For the other figures that were not based on your mass spec data, we would not require all raw data. Rather, we ask that all individual quantitative observations that underlie the data summarized in the figures and results of your paper be made available in one of the following forms:

a. Supplementary files (e.g., excel). Please ensure that all data files are uploaded as 'Supporting Information' and are invariably referred to (in the manuscript, figure legends, and the Description field when uploading your files) using the following format verbatim: S1 Data, S2 Data, etc. Multiple panels of a single or even several figures can be included as multiple sheets in one excel file that is saved using exactly the following convention: S1_Data.xlsx (using an underscore).

b. Deposition in a publicly available repository. Please also provide the accession code or a reviewer link so that we may view your data before publication.

>>Regardless of the method selected, please ensure that you provide the individual numerical values that underlie the summary data displayed in any figure that is not presenting mass spec data (for example figures with qPCR or western blot analyses)

>>Please also ensure that figure legends in your manuscript include information on where the underlying data can be found, and ensure your supplemental data file/s has a legend.

>>Please ensure that your Data Statement in the submission system accurately describes where your data can be found.

9) WESTERN BLOTS: Thank you for providing a supplemental file with the underlying western blot images for your study. Some of the images provided in that file look a bit cropped to me. Please provide a fully uncropped version for all blots. Ideally we should either be able to see the edges of the membrane, or you can provide the full image taken from the scanner.

10) CODE: Per journal policy, if you have generated any custom code during the course of this investigation, please make it available without restrictions. Please ensure that the code is sufficiently well documented and reusable, and that your Data Statement in the Editorial Manager submission system accurately describes where your code can be found. [

We expect to receive your revised manuscript within two weeks.

*Published Peer Review History*

*Press*

Sincerely,

Luke

Lucas Smith, Ph.D.

Senior Editor

lsmith@plos.org

PLOS Biology

Reviewer remarks:

Reviewer #2: The authors have thoroughly addressed all my questions and concerns. I commend their thoughtful and comprehensive responses.

---

## [Editor Report · Decision Letter 3]

19 Dec 2025

Dear Meilian,

Thank you for the submission of your revised Research Article "Periodic Fasting and Refeeding Re-shapes Lipid Saturation, Storage, and Distribution in Brown Adipose Tissue" for publication in PLOS Biology and for addressing our last editorial requests in this revision. On behalf of my colleagues and the Academic Editor, Hoon-Ki Sung, I am pleased to say that we can in principle accept your manuscript for publication, provided you address any remaining formatting and reporting issues. These will be detailed in an email you should receive within 2-3 business days from our colleagues in the journal operations team; no action is required from you until then. Please note that we will not be able to formally accept your manuscript and schedule it for publication until you have completed any requested changes.

**IMPORTANT - A few last editorial notes:

1) Thank you again for providing the raw mass spec data for your study on the MassIVE dataset repository. Please note, that I have taken the liberty of updating your Data Availability Statement in our online system to reference this dataset. The new statement reads:

"The raw metabolomics, lipidomics, and MALDI MSI data are available in MASSIVE datasets with identifier MSV000100122. All relevant data are within the paper and its Supporting Information files"

^Please take a moment to double check that this looks OK. Note the version in our editorial manager system is teh one that will be published with your study, so you can remove the data statement that is in your manuscript. While it is great to have the raw data for those studies, please also leave the processed tables that you have already provided in the supplemental tables as well.

2) As discussed, I noticed that several of your supplemental figures are two pages and I am not sure that is allowed? I will flag this to production for their thoughts, but you may need to either shrink those figures into 1 page, or split them into two figures each. Either way is fine with me, and I will leave it up to you and production to decide which is the best way forward.

PRESS

Sincerely,

Luke

Lucas Smith, Ph.D.

Senior Editor

PLOS Biology

lsmith@plos.org